# On Joint Regularization and Calibration in Deep Ensembles

**Laurits Fredsgaard**                                            *laula@dtu.dk*
*Department of Applied Mathematics and Computer Science*
*Technical University of Denmark*

**Mikkel N. Schmidt**                                            *mnsc@dtu.dk*
*Department of Applied Mathematics and Computer Science*
*Technical University of Denmark*

**Reviewed on OpenReview:** *https://openreview.net/forum?id=6xqV7DP3Ep*

## Abstract

Deep ensembles are a powerful tool in machine learning, improving both model performance and uncertainty calibration. While ensembles are typically formed by training and tuning models individually, evidence suggests that jointly tuning the ensemble can lead to better performance. This paper investigates the impact of jointly tuning weight decay, temperature scaling, and early stopping on both predictive performance and uncertainty quantification. Additionally, we propose a partially overlapping holdout strategy as a practical compromise between enabling joint evaluation and maximizing the use of data for training. Our results demonstrate that jointly tuning the ensemble generally matches or improves performance, with significant variation in effect size across different tasks and metrics. We highlight the trade-offs between individual and joint optimization in deep ensemble training, with the overlapping holdout strategy offering an attractive practical solution. We believe our findings provide valuable insights and guidance for practitioners looking to optimize deep ensemble models. Code is available at: `https://github.com/lauritsf/ensemble-optimality-gap`

## 1 Introduction

Deep ensembles are a simple and practical method that combines multiple independently trained models to enhance predictive accuracy, improve robustness, and provide uncertainty estimates (Lakshminarayanan et al., 2017). Their effectiveness relies on having diverse members that have uncorrelated errors, which reduces variance and minimizes the impact of individual model mistakes (Hansen & Salamon, 1990; Krogh & Sollich, 1997).

While individual models in an ensemble may differ in architecture, training set, and other factors, a common practice is to train ensembles using the same model architecture, where the only differences are the initializations and the order in which the training examples are presented. This also offers a simple and effective method for selecting regularization hyperparameters such as weight decay and dropout: These settings can be optimized for a single model, typically through grid search, and then used to train the ensemble members independently. Similarly, if post-hoc calibration or early stopping is used, it is often applied to each ensemble member independently.

This approach simplifies the tuning process, but while an ensemble of well-regularized and well-calibrated models will generally perform well, it may not be the optimal strategy. This potential mismatch, which we term the *ensemble optimality gap*, arises because what is optimal for a single model is not necessarily optimal for the final ensemble. The theoretical basis for this is that the expected loss of an ensemble is fundamentally different from the average loss of its members, often involving a beneficial diversity term (Wood et al., 2023; Krogh & Sollich, 1997). Consequently, using a single model's validation performance as a proxy for the final ensemble's test performance creates a generalization mismatch. This flawed validation process prevents the discovery of hyperparameters that might make beneficial trade-offs, such as sacrificing marginal individual

model performance for a larger gain in diversity. While other research focuses on explicitly inducing diversity, our work investigates a simpler premise: closing the optimality gap by ensuring the validation objective correctly mirrors the final deployment objective—the ensemble itself.

Previous work has shown that allowing individual models within an ensemble to overfit to a certain extent can lead to improvements in both prediction accuracy (Sollich & Krogh, 1995) and calibration (Wu & Gales, 2021). In practice, however, this is often disregarded because tuning the complete ensemble by holdout or cross-validation can be a considerable computational expense or does not seamlessly fit into existing workflows. Hyperparameter tuning (such as grid search) requires training an entire ensemble for each parameter combination, scaling the computational cost with the ensemble size. However, methods like early stopping can be validated during parallel training with minimal added cost, whereas post hoc techniques like temperature scaling can be evaluated on the ensemble without additional expense.

In this paper, we systematically explore the ensemble optimality gap across three key aspects of deep ensemble training and calibration: weight decay tuning, temperature scaling, and early stopping. Our objective is to assess the magnitude of this effect in common settings and demonstrate how it can be mitigated by optimizing for ensemble performance. In particular, we examine how the optimality gap influences model accuracy, uncertainty calibration, and predictive likelihood, as well as investigate its impact on ensemble diversity.

To enhance ensemble diversity, a well-known approach is to train the ensemble using a k-fold cross-validation strategy, where each ensemble member is validated on separate holdout data. While this approach can improve the estimation of generalization error for a single model, it prevents direct validation of the full ensemble performance, as a common validation data must be held out for all ensemble members. This leads to a choice between increasing ensemble diversity and having the ability to tune the ensemble as a whole. Both strategies have been demonstrated to lead to improved performance. We explore a strategy that balances these factors by using partially overlapping holdout sets across ensemble members.

Finally, when training standard deep ensembles is impractical, techniques like batch ensembles or multiple-input multiple-output (MIMO) ensembles offer viable alternatives. In these approaches, the ensemble is formed by *sub-models* with partially shared parameters within a single, larger model that is trained in one run. We demonstrate how our overlapping holdout strategy can be applied in the batch ensemble setting and compare its performance across different initialization strategies.

In summary, our work addresses the following aspects:

- We demonstrate several settings in which the *ensemble optimality gap* is significant, and show to which extent it can be mitigated by jointly tuning the ensemble.

- We propose a novel *overlapping holdout* validation strategy that sits between using a common shared holdout set and using independent holdouts as in k-fold cross-validation.

- We present a case study based on *batch ensembling* that demonstrates how an ensemble can be jointly trained and tuned in a single run with the overlapping holdout validation strategy.

We validate our study empirically using four diverse benchmark tasks spanning image, graph, tabular, and text classification. Our results demonstrate clear benefits from validating the ensemble jointly, especially for early stopping and temperature scaling, compared to validating individual models, while joint weight decay tuning shows more nuanced effects predominantly related to calibration. We assess the utility of the overlapping holdout strategy in different settings and also provide key insights regarding initialization choices for efficient batch ensembles. Collectively, these findings offer concrete guidance for practitioners on navigating the trade-offs between individual and joint optimization when training and calibrating deep ensembles.

## 2 Background

### 2.1 Ensemble Methods: Foundations and Diversity

Ensemble methods improve prediction and robustness by combining multiple models (Dietterich, 2000). The core principle relies on combining outputs from diverse members with uncorrelated errors, thus reducing variance and improving generalization (Hansen & Salamon, 1990; Krogh & Sollich, 1997). In this paper, we focus on deep ensembles (Lakshminarayanan et al., 2017), a simple and effective technique for deep neural networks. For classification tasks, we consider the common approach where the ensemble prediction is the arithmetic mean of the softmax probabilities from individual members,

$$\bar{\boldsymbol{p}}(y|x) = \frac{1}{M} \sum_{m=1}^{M} \boldsymbol{p}(y|x, \theta_m), \tag{1}$$

corresponding to a uniform mixture over the ensemble members (see e.g. Tassi et al. (2022) for a discussion of the pros and cons of this strategy).

While effective, standard deep ensembles incur substantial costs, as training, storing, and running $M$ independent models leads roughly to an $M$-fold increase in computation time at training and inference. This scalability challenge has spurred the development of more efficient ensemble methods. These approaches often reduce the computational or parameter costs through techniques like parameter sharing (e.g., BatchEnsemble (Wen et al., 2020b)), creating implicit ensembles (e.g., MIMO (Havasi et al., 2021), Early Exits (Qendro et al., 2021)), leveraging stochastic inference (e.g., MC Dropout (Gal & Ghahramani, 2016)), or developing efficient Bayesian approximations (e.g., Rank-1 BNNs (Dusenberry et al., 2020)).

**Implicit and Explicit Diversity.** Standard Deep Ensembles typically use identical architectures trained independently. Diversity is achieved *implicitly*, primarily through different random weight initializations, which serve as the main source of functional diversity, although using distinct stochastic batches during training also contributes to a lesser extent (Fort et al., 2020). Data resampling techniques such as Bagging (Breiman, 1996) can promote beneficial diversity for traditional models but are often detrimental for deep networks (Lee et al., 2015; Lakshminarayanan et al., 2017). This is largely because deep models are sensitive to training data size, and the reduction in data per bagged member significantly weakens the individual predictors. Furthermore, regularization techniques applied during training (such as weight decay or early stopping), while improving individual model generalization, may also implicitly constrain the diversity among ensemble members. Although this implicit diversity (influenced by initialization, data splits, stochastic batches, and regularization) is often sufficient, explicit diversity-enhancing techniques can also lead to improvements, e.g., by modifying training losses or adding regularization (see e.g., Liu & Yao, 1999; Pagliardini et al., 2023; Jain et al., 2020), but their necessity and benefit, especially for large models, is debated (Abe et al., 2022; 2024). In some cases, joint training methods can lead to *learner collusion* (Jeffares et al., 2023), a phenomenon where the ensemble members increase their diversity in a way that does not improve generalization.

**Quantifying Diversity.** Quantifying the diversity among ensemble members provides key insights into their collective behavior and prediction characteristics. For probabilistic predictive models, a useful information-theoretic metric for ensemble diversity is the difference between the entropy of the average predictive distribution, $\bar{\boldsymbol{p}}_i$, and the average entropy of individual member predictions, $\boldsymbol{p}_i^m$. This is equivalent to the average KL divergence $D_{\text{KL}}(\boldsymbol{p}_i^m || \bar{\boldsymbol{p}}_i)$ (see Appendix B for details):

$$D_i = H(\bar{\boldsymbol{p}}_i) - \frac{1}{M} \sum_{m=1}^{M} H(\boldsymbol{p}_i^m). \tag{2}$$

where $H$ denotes the Shannon entropy. In the classification setting with $\bar{\boldsymbol{p}}_i$ defined as the geometric mean, this expression has a natural interpretation in the form of a bias, variance, diversity decomposition of the

expected loss (Wood et al., 2023); however, in this work, we use the arithmetic mean, as it is more commonly applied. For an overview of alternative diversity metrics, see, e.g., Kuncheva & Whitaker (2003); Heidemann et al. (2021).

## 2.2 Calibration of Deep Learning Models and Ensembles

Beyond predictive accuracy, the reliability of a model's confidence estimates is crucial for dependable decision-making, particularly in risk-sensitive applications (Niculescu-Mizil & Caruana, 2005). A model is considered well-calibrated if its predicted probabilities accurately reflect the true likelihood of correctness (e.g., predictions made with 80% confidence are correct 80% of the time). While modern deep neural networks achieve high accuracy, they are often found to be poorly calibrated, typically exhibiting overconfidence in their predictions (Guo et al., 2017).

Calibration is commonly evaluated using metrics such as the expected calibration error (ECE), which measures the discrepancy between confidence and accuracy across prediction bins (Naeini et al., 2015), and the negative log-likelihood (NLL) of the true classes. NLL is a proper scoring rule, meaning it is uniquely minimized when predicted probabilities match the true underlying probabilities, thus rewarding both accuracy and calibration (Gneiting & Raftery, 2007).

**Temperature Scaling.**   A simple yet effective post-hoc technique for improving calibration is temperature scaling (Guo et al., 2017). It involves rescaling the model's output logits $\boldsymbol{z}(\boldsymbol{x})$ by a single positive scalar parameter, the temperature $T$, before applying the softmax function according to the formula

$$\boldsymbol{p}(\boldsymbol{x};T) = \mathrm{softmax}\left(\frac{\boldsymbol{z}(\boldsymbol{x})}{T}\right). \tag{3}$$

A temperature $T > 1$ softens the probability distribution (increasing entropy, reducing confidence), while $T < 1$ sharpens it. The optimal $T$ is typically found as the value that minimizes some calibration metric on a held-out validation dataset $\mathcal{D}_{\mathrm{val}}$. Using the NLL as the metric, the optimal temperature is given by

$$\underset{T>0}{\arg\min} \sum_{(\boldsymbol{x}_j,y_j)\in\mathcal{D}_{\mathrm{val}}} -\log \boldsymbol{p}(\boldsymbol{x}_j;T)_{y_j} \tag{4}$$

where $\boldsymbol{p}(\boldsymbol{x}_j;T)_{y_j}$ denotes the predicted probability for the true class $y_j$. Since $T$ only rescales logits before the softmax, it does not change individual models' accuracies.

**Individual vs. Joint calibration**   When applying temperature scaling to an ensemble of $M$ models, two main strategies arise, differing primarily in how the temperature parameter(s) are optimized and applied. It is important to note that *any* strategy involving temperature scaling applied before averaging the outputs of the non-linear softmax function can potentially alter the final classification outcome (i.e., the $\arg\max$ of the averaged probabilities) compared to averaging unscaled probabilities. The two main implementation strategies are:

- **Individual Temperature Scaling:** A separate temperature $T_m$ is optimized for each ensemble member $m$, typically using its own validation set $\mathcal{D}_{\mathrm{val}}^{(m)}$. The final ensemble prediction is the average of these individually calibrated probability vectors, given by

$$\bar{\boldsymbol{p}}^{\mathrm{individual}}(\boldsymbol{x}) = \frac{1}{M}\sum_{m=1}^{M} \mathrm{softmax}\left(\frac{\boldsymbol{z}_m(\boldsymbol{x})}{T_m}\right). \tag{5}$$

  Here, the potential impact on the classification outcome is influenced by the use of different scaling factors $T_m$ across members.

- **Joint Temperature Scaling:** A single, shared temperature $T_{\text{joint}}$ is optimized for the entire ensemble using a suitable joint validation set $\mathcal{D}_{\text{val}}^{\text{joint}}$. This shared temperature $T_{\text{joint}}$ is applied to the logits $\boldsymbol{z}_m(\boldsymbol{x})$ of each member before the softmax activation, and the resulting calibrated probabilities are then averaged according to the formula

$$\bar{\boldsymbol{p}}^{\text{joint}}(\boldsymbol{x}) = \frac{1}{M} \sum_{m=1}^{M} \text{softmax}\left(\frac{\boldsymbol{z}_m(\boldsymbol{x})}{T_{\text{joint}}}\right). \tag{6}$$

  Although the scaling $T_{\text{joint}}$ is uniformly applied (preserving the $\arg\max$ of individual members), the ensemble class prediction can change as averaging happens after the probability distributions are tempered. The optimal $T_{\text{joint}}$ is found by minimizing a calibration metric (such as NLL) of this final averaged prediction $\bar{\boldsymbol{p}}^{\text{joint}}(\boldsymbol{x})$ on a joint validation set $\mathcal{D}_{\text{val}}^{\text{joint}}$.

A relevant alternative is the *Pool-Then-Calibrate* strategy, where temperature is applied after averaging probabilities, which preserves the final prediction (Rahaman & Thiéry, 2021). This contrasts with our method of scaling logits before averaging, which allows for diagnostic checks on individual members. Despite these conceptual differences, our results confirm the findings of Rahaman & Thiéry (2021) that both joint scaling methods perform comparably in practice (see Appendix D for a comparison).

The relationship between individual member calibration and overall ensemble calibration is non-trivial. Importantly, Wu & Gales (2021) showed that ensembling individually calibrated models does not guarantee a well-calibrated ensemble and can lead to under-confidence, advocating instead for calibration strategies that consider the ensemble effect. Their work focused specifically on optimizing temperature scaling parameters by minimizing ECE, whereas optimization based on proper scoring rules like NLL represents an alternative calibration objective commonly used for model training and evaluation.

### 2.3 Hyperparameter Tuning for Ensembles

Selecting appropriate hyperparameters, such as regularization strengths (e.g., weight decay) or learning parameters, is critical for training performant deep learning models. For ensembles, this presents a fundamental choice regarding the optimization objective: Should hyperparameters be tuned to optimize the performance of individual members, or the performance of the ensemble as a whole?

A common practice, largely due to simplicity and significantly lower computational cost, involves tuning hyperparameters for a single model (e.g., via grid search or random search (Bergstra & Bengio, 2012)) and then applying the selected configuration uniformly to all ensemble members during their independent training (Lakshminarayanan et al., 2017). Directly tuning for the ensemble objective, by contrast, would necessitate training and evaluating the *entire* ensemble for *each* candidate hyperparameter setting, incurring a computational cost that typically scales with the ensemble size and is often prohibitively expensive.

Beyond searching for a single optimal setting to apply uniformly, alternative strategies exist that leverage the models generated during hyperparameter exploration or explicitly use hyperparameter diversity. For instance, Wenzel et al. (2020) proposed *hyper-deep ensembles*, a method that explicitly combines models resulting from different hyperparameter settings (found via random search and greedy selection) and different random weight initializations. This combination of diversity sources was shown to improve robustness and uncertainty quantification compared to ensembles relying solely on random initialization. Similarly, Jin & Wu (2024) construct ensembles from models saved during learning rate schedule tuning runs, arguing this efficiently reuses computational effort and enhances diversity, reporting strong performance. A limitation of directly using models from tuning runs, however, is that the validation data used for hyperparameter selection is not incorporated into the training data for the final models, differing from standard workflows where models are typically retrained on combined data after tuning.

While these alternative construction methods exist, the common practice remains to tune a single configuration for standard deep ensembles. This standard practice implicitly assumes that hyperparameters optimal for a single model are also (close to) optimal for the ensemble, which might not hold true in practice, an

issue also noted by Gorishniy et al. (2025). However, there appears to be limited research directly comparing individual versus ensemble-based hyperparameter tuning for standard deep ensembles.

### 2.4 Early Stopping Ensembles

Early stopping is another widely used regularization technique that prevents overfitting by monitoring performance on a validation set and terminating training when performance on this set ceases to improve (Prechelt, 1998). Alternatively, the stopping point can be guided by estimators of generalization error that do not require held-out data, such as those derived from bootstrap ensembles (Hansen et al., 1997). When applied to ensembles, the main approaches are:

- **Individual Early Stopping:** Each ensemble member $m$ is trained and stopped independently based on its own performance, monitored on a validation set $\mathcal{D}_{val}^{(m)}$. Training for member $m$ halts when its validation metric fails to improve for a predefined patience period.

- **Joint Early Stopping:** The performance of the *entire ensemble* (e.g., NLL of the average prediction) is monitored on a joint validation set $\mathcal{D}_{val}^{joint}$. Training for *all* members stops simultaneously when the *ensemble's* performance has not improved for the specified patience.

While individual early stopping optimizes each member in isolation, early ensemble theory suggests that allowing individual members to overfit slightly may be beneficial for the final ensemble performance (Sollich & Krogh, 1995). Joint early stopping might naturally facilitate this by potentially allowing longer training compared to the point where individual models start overfitting on their respective validation sets. Despite its conceptual appeal, the comparative benefits of individual versus joint stopping strategies for modern deep ensembles appear less explored. The feasibility and specific mechanism for evaluating ensemble performance for joint early stopping depend critically on the chosen validation data strategy (Section 3.3). Importantly, unlike potentially costly joint hyperparameter grid searches, joint early stopping can often be implemented with minimal computational overhead compared to individual early stopping, especially when members are trained in parallel.

## 3 Methodology

This section details the methodology used to empirically evaluate the *ensemble optimality gap*. We begin by formally defining the problem and then describe the specific experimental settings used across our study.

### 3.1 Formalizing the Ensemble Optimality Gap

Our investigation centers on comparing individual and joint optimization strategies. This comparison can be framed as a general problem of model selection, where the goal is to select the best hypothesis, $h$, from a set of candidate hypotheses, $\mathcal{H}_{cand}$, by evaluating a loss function, $\mathcal{L}$, on a validation set, $\mathcal{D}_{val}$. A hypothesis $h$ can represent a specific hyperparameter configuration, such as a weight decay value, a temperature for calibration, or a training epoch for early stopping.

The core of this issue lies in the validation objective. Using a single model's validation performance as a proxy for the final ensemble's performance creates a mismatch: The quantity being optimized (single-model loss) does not align with the ultimate deployment goal (ensemble loss). This discrepancy—which we term the *ensemble optimality gap*—arises because the loss of an ensemble's prediction is not merely the average of its members' individual losses.

This relationship is formally captured by the following decomposition.

**Ambiguity Decomposition**  Assuming a convex loss, such as squared error or cross-entropy, the ensemble loss can be decomposed into two parts—the average individual model loss and an ambiguity term:

$$\underbrace{\mathcal{L}\left(\frac{1}{M}\sum_{m=1}^{M}f_m\right)}_{\text{Ensemble Loss}} = \underbrace{\frac{1}{M}\sum_{m=1}^{M}\mathcal{L}(f_m)}_{\text{Average Loss}} - \text{Ambiguity} \tag{7}$$

The Ambiguity term quantifies the beneficial diversity within the ensemble, and its exact form depends on the loss. For instance, with the squared error loss, the Ambiguity term is the average squared difference of each model's prediction from the ensemble's mean prediction, directly measuring their disagreement. For cross-entropy loss with an arithmetic mean combiner, the term is more complex and becomes dependent on the true label, capturing the divergence between the arithmetic and geometric means of the model probabilities (Wood et al., 2023). The Ambiguity is always non-negative, which follows from Jensen's inequality.

We have a model $f_{\theta,h}$ with trainable parameters $\theta$ and hyperparameters $h$ which define conditions such as weight decay, temperature scaling, and stopping epoch.

Given the hyperparameters, $h$, optimal model parameters are found by minimizing the loss on training data, e.g. using stochastic gradient descent.

$$\theta^*(h) = \arg\min_{\theta}\mathcal{L}_{\text{train}}(f_{\theta,h}) \tag{8}$$

and the optimal hyperparameters are chosen by minimizing a validation loss,

$$h^* = \arg\min_{h\in\mathcal{H}}\mathcal{L}_{\text{val}}(f_{\theta^*(h),h}) \tag{9}$$

In practice, weight decay is typically tuned using grid search; temperature scaling may use either grid search or convex optimization; and early stopping is usually implemented as a greedy procedure that halts training when the validation error no longer improves, possibly after a patience period.

To make the following specific ensemble validation strategies easier to parse, we will use a convenient shorthand. We denote the fully trained $m$-th model with hyperparameters $h$ as $f_m(h) \equiv f_{\theta_m^*(h),h}$.

**The Optimality Gap** The optimality gap emerges from how we choose to define the validation loss when searching for the optimal hyperparameter $h^*$. There are two distinct strategies based on the decomposition:

1. **Joint validation** directly evaluates the final objective of interest: the performance of the entire ensemble. It finds $h_{\text{ens}}^*$ by using the complete ensemble loss for validation, thereby accounting for both the average performance of the models and the ambiguity between them:

$$h_{\text{ens}}^* = \arg\min_{h\in\mathcal{H}}\mathcal{L}_{\text{val}}\left(\frac{1}{M}\sum_{m=1}^{M}f_m(h)\right) \tag{10}$$

2. **Individual validation** uses a practical but incomplete proxy. It finds $h_{\text{ind}}^*$ by using the average individual loss (or a single model's loss) for validation. This common approach simplifies the optimization but is structurally blind to the ambiguity term:

$$h_{\text{ind}}^* = \arg\min_{h\in\mathcal{H}}\frac{1}{M}\sum_{m=1}^{M}\mathcal{L}_{\text{val}}(f_m(h)) \tag{11}$$

Since these two strategies optimize different objectives–one the complete objective and the other a simplified proxy–they generally yield different hyperparameters ($h_{\text{ind}}^* \neq h_{\text{ens}}^*$). This may lead to a measurable performance difference on the final test set.

We formally define the *ensemble optimality gap* as the difference in the test loss between the ensemble tuned with the individual-loss proxy and the one tuned with the joint objective:

$$\text{Optimality Gap} = \mathcal{L}_{\text{test}}\left(\frac{1}{M}\sum_{m=1}^{M}f_m(h_{\text{ind}}^*)\right) - \mathcal{L}_{\text{test}}\left(\frac{1}{M}\sum_{m=1}^{M}f_m(h_{\text{ens}}^*)\right). \tag{12}$$

A positive gap demonstrates a tangible benefit to the joint strategies. Our experiments are designed to empirically measure the magnitude of this gap under various conditions.

## 3.2 Datasets and Base Models

To assess the generality of our findings on ensemble optimization, we conducted experiments across four distinct and commonly used benchmark domains, differing significantly in data modality, task complexity, data scale, and model size:

- **Image Classification** (CIFAR-10 / WRN-16-4): Our first domain uses the CIFAR-10 dataset (Krizhevsky, 2009), a widely-used 10-class image classification benchmark with 50,000 training images. We pair this with a Wide ResNet-16-4 (WRN-16-4) model (Zagoruyko & Komodakis, 2016), a common CNN architecture for this task containing approximately 2.7 million parameters. This combination represents a setting where a high-capacity model operates on a moderately sized dataset, suggesting a significantly overparameterized regime. This characteristic makes it particularly suitable for studying the interplay between ensemble methods and factors like regularization (e.g., weight decay, early stopping) and multi-class calibration.

- **Graph Classification** (NCI1 / GCN): As a contrasting setting with structured data, we used the NCI1 graph classification benchmark (Shervashidze et al., 2011; Wale et al., 2008), a binary classification task with significantly fewer data points (4,110 graphs total, 3,288 used for training), with a four-layer Graph Convolutional Network (GCN). This pairing allows us to test our hypotheses on a different data modality and an architecture with substantially fewer parameters (24,204). Despite the vastly different scale, regularization remains important for the GCN's generalization on this dataset, allowing us to examine ensemble optimization effects in a different modeling context. Furthermore, the limited data availability may emphasize the impact of data allocation in different validation holdout strategies.

- **Tabular Classification** (Covertype / MLP): To cover a third modality and explore larger ensemble sizes, we include the Covertype dataset (Blackard & Dean, 1999) from the UCI repository. This is a large-scale tabular benchmark with approximately 581,012 samples, 54 features, and 7 classes. We use a three-hidden-layer Multilayer Perceptron (MLP) and an ensemble size of $M = 8$ for this task. This setting allows us to test our conclusions in a high-data, non-convolutional regime and investigate trends with a larger number of ensemble members.

- **Text Classification** (AG News / BiLSTM): Our final domain involves text classification using the AG News dataset (Zhang et al., 2015), which consists of 120,000 training samples and 7,600 test samples across 4 balanced classes. We employ a Bidirectional Long Short-Term Memory (BiLSTM) network (Hochreiter & Schmidhuber, 1997; Schuster & Paliwal, 1997) which processes inputs prepared by the GPT-2 tokenizer (Radford et al., 2019). This task introduces a sequential data modality and allows us to evaluate our findings on a common natural language processing architecture with an ensemble size of M=8.

## 3.3 Validation Data Strategies for Ensemble Evaluation

How validation data is assigned to ensemble members impacts training and evaluation, particularly when considering joint ensemble objectives versus individual member training needs. We define $\mathcal{D}'$ as the available data excluding the final test set, $\mathcal{D}_{\text{val}}^{(m)}$ as the validation set for member $m$, and $\mathcal{D}_{\text{train}}^{(m)}$ as its training set. We consider three primary strategies for an ensemble of size $M$:

- **Shared Holdout:** All members use the same split: $\mathcal{D}_{\text{train}}^{(m)} = \mathcal{D}_{\text{train}}$ and $\mathcal{D}_{\text{val}}^{(m)} = \mathcal{D}_{\text{val}}$ for all $m$. $\mathcal{D}_{\text{val}}$ is not used by any member during training. This allows direct evaluation of the full ensemble on $\mathcal{D}_{\text{val}}$.

- **Disjoint Holdout:** Each ensemble member $m$ uses its own unique validation set, $\mathcal{D}_{\text{val}}^{(m)} \subset \mathcal{D}'$, of a predefined size. These validation sets are mutually disjoint, and their combined size cannot exceed the total available data $|\mathcal{D}'|$. Since member $m$ trains on all data except its own validation set (i.e., $\mathcal{D}_{\text{train}}^{(m)} = \mathcal{D}' \setminus \mathcal{D}_{\text{val}}^{(m)}$), the data points used to validate it are necessarily included in the training data for all other members, maximizing training data utilization across the ensemble. However, this structure prevents direct evaluation of the full ensemble.

- **Overlapping Holdout:** Each model $m$ gets a unique validation set $\mathcal{D}_{\text{val}}^{(m)}$ composed of two distinct parts (*halves*). Each half is shared with one neighboring model in a cyclical manner. For instance, half of Model 2's validation data is shared with Model 1, the other half with Model 3. This allows pairwise joint evaluation on the shared halves. (Formally, using $M$ data portions $S_k$ that partition $\mathcal{D}'$, then $\mathcal{D}_{\text{val}}^{(m)} = S_m \cup S_{m+1 \pmod M}$ and $\mathcal{D}_{\text{train}}^{(m)} = \mathcal{D}' \setminus \mathcal{D}_{\text{val}}^{(m)}$). While this strategy does not permit estimating the validation performance of the entire ensemble on held-out data, it allows joint estimation of model pairs as an approximation.

Choosing among these strategies involves balancing the trade-off between maximizing data utilization for training and enabling joint evaluation of the ensemble. The disjoint holdout strategy maximizes data usage but precludes any joint evaluation. Conversely, the shared holdout strategy enables direct, full-ensemble evaluation but at the cost of reserving a portion of data that is never used for training by any model. The overlapping holdout strategy offers a practical compromise. While it does not permit evaluating the entire ensemble, it allows for pairwise joint evaluation. Given that the most substantial gains in ensemble performance are often observed when moving from a single model to an ensemble of two, this pairwise evaluation can serve as a proxy for full ensemble performance, making it a potentially useful option in data-scarce settings.

## 4 Experiments

We conduct a series of experiments to empirically measure the ensemble optimality gap across three key areas: hyperparameter tuning, temperature scaling, and early stopping.

### 4.1 Hyperparameter Tuning (Weight Decay)

**Purpose**   This experiment aims to investigate how model performance is affected by different strategies for tuning the weight decay hyperparameter. Specifically, we compare two approaches: optimizing weight decay for a single model, and optimizing it to maximize the full ensemble's performance. This comparison aims to quantify the ensemble optimality gap and shed light on whether tuning for the ensemble yields better results.

**Method**   Optimal weight decay was determined via grid search, minimizing validation NLL using a shared holdout set. We compared selecting the best value based on the average individual model NLL versus the NLL of the ensemble's average prediction. Models were trained using stochastic gradient descent (SGD) with momentum and cosine annealing (experimental parameters are detailed in Appendix A.3).

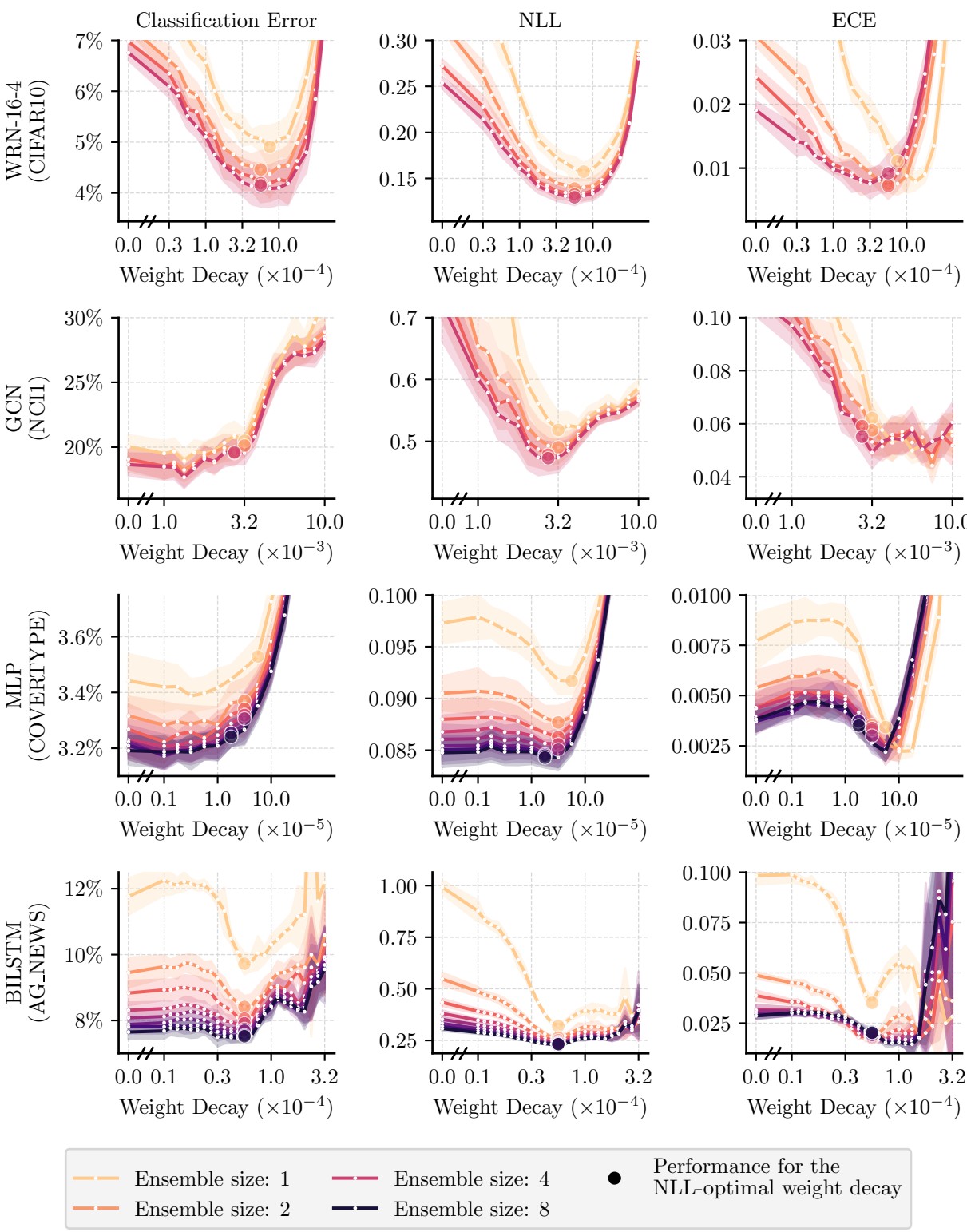

Figure 1: Validation performance across varying weight decay values for a WRN-16-4 on CIFAR-10, a GCN on NCI1, an MLP on Covertype, and a BiLSTM on AG News. The plots show results for ensemble sizes 1 to 4 (WRN and GCN) and 1 to 8 (MLP and BiLSTM). The optimal weight decay for each ensemble size is selected based on the lowest average NLL. (WRN: Wide ResNet; GCN: Graph Convolutional Network; MLP: Multi-Layer Perceptron; BiLSTM: Bidirectional Long Short-Term Memory; NLL: negative log-likelihood; ECE: expected calibration error).

Table 1: Test performance of the full ensemble, where optimal weight decay (WD*) was tuned by minimizing validation NLL. Values in **bold** are not statistically different from the best result (within 1.96 SEM). The ensemble size is 4 for WRN-16-4 and GCN, and 8 for MLP and BiLSTM.

| Model / Dataset | WD* | Metric | Single-Model Opt. Ensemble | Ensemble Opt. Ensemble |
|---|---|---|---|---|
| WRN-16-4 CIFAR10 | S: 7.50e-04 E: 5.62e-04 | Err. (%) ↓ NLL ↓ ECE ↓ | **4.09** $_{\pm 0.07}$ **0.129** $_{\pm 0.000}$ 0.009 $_{\pm 0.000}$ | **4.12** $_{\pm 0.04}$ **0.127** $_{\pm 0.001}$ **0.006** $_{\pm 0.000}$ |
| GCN NCI1 | S: 3.16e-03 E: 2.74e-03 | Err. (%) ↓ NLL ↓ ECE ↓ | 19.22 $_{\pm 0.16}$ 0.443 $_{\pm 0.002}$ **0.034** $_{\pm 0.002}$ | **18.47** $_{\pm 0.13}$ **0.435** $_{\pm 0.002}$ **0.034** $_{\pm 0.001}$ |
| MLP COVERTYPE | S: 5.62e-05 E: 1.78e-05 | Err. (%) ↓ NLL ↓ ECE ↓ | 3.31 $_{\pm 0.01}$ 0.084 $_{\pm 0.000}$ **0.001** $_{\pm 0.000}$ | **3.16** $_{\pm 0.02}$ **0.083** $_{\pm 0.000}$ 0.004 $_{\pm 0.000}$ |
| BILSTM AG_NEWS | S: 5.62e-05 E: 5.62e-05 | Err. (%) ↓ NLL ↓ ECE ↓ | **7.55** $_{\pm 0.06}$ **0.232** $_{\pm 0.002}$ **0.020** $_{\pm 0.001}$ | **7.55** $_{\pm 0.06}$ **0.232** $_{\pm 0.002}$ **0.020** $_{\pm 0.001}$ |

**Results**   The validation sweeps in Figure 1 show that for the WRN, GCN, and MLP models, the optimal weight decay (in terms of NLL) tends to shift towards less regularization as the ensemble size increases. This trend is consistent with the theory that allowing individual members more flexibility can benefit the collective. For the BiLSTM, however, the optimal value remains consistent regardless of ensemble size.

The final test performance, summarized in Table 1, reveals a nuanced picture of the trade-offs between the tuning strategies. Optimizing for the ensemble's NLL is a principled approach that can yield targeted improvements. For instance, it results in the best calibration (ECE) for WRN and improves the classification error for the GCN and MLP compared to the single-model optimum. However, this strategy does not uniformly outperform the others. The final ensemble consistently surpasses the performance of its constituent parts in all settings. This robustness is highlighted by the trend towards less regularization (Figure 1) and is most stark when no weight decay is used at all (Appendix Table 4). We include the average individual model performance in Appendix Table 3 as a diagnostic metric; this provides context on how changes to the ensemble's behavior relate to the performance of its underlying members.

**Conclusions**   Our findings suggest that while tuning weight decay on a single model provides a strong baseline, it can be a suboptimal proxy for the final ensemble's performance. Optimizing on the full ensemble's validation objective offers a path to improved results, but these benefits are modest and not guaranteed across all metrics.

The gains from ensemble-level tuning, when present, are often specific, improving either classification error (GCN) or calibration (WRN) depending on the context. Practitioners must therefore weigh the potential for a targeted performance boost against the increased computational cost of tuning the entire ensemble. Crucially, the choice of tuning metric shapes the outcome; optimizing for NLL, as we do here, directly improves that metric but does not guarantee the best ECE, exposing a clear trade-off between a model's predictive accuracy and its calibration.

## 4.2 Temperature Scaling for Calibration

**Purpose**   This experiment investigates how different temperature scaling strategies impact the calibration and overall performance of deep ensembles. We specifically compare optimizing temperature(s) for individual models versus optimizing a single temperature for joint ensemble prediction, evaluating these approaches under different validation holdout strategies and varying validation set sizes.

**Method**   Base models were first trained using parameters from single-model weight decay tuning. Subsequently, to compare individual and joint temperature scaling strategies across shared and overlapping validation holdouts, we optimized the temperature(s) using L-BFGS to minimize validation NLL (experimental parameters are detailed in Appendix A.3).

**Results**   The results, presented in Figure 2, reveal a clear trade-off related to the validation set size across all four experimental settings: using too little data can lead to unstable temperature estimates, while using too much reduces the available training data and degrades overall performance. This trade-off is modulated by the holdout strategy. The overlapping holdout strategy demonstrates greater robustness to large validation percentages compared to the shared holdout, particularly for the GCN and MLP models. This is likely due to its more efficient use of data for training. However, for most configurations, using a smaller validation set (e.g., 1–5%) with the shared holdout strategy yields the most competitive results.

Regarding the scaling methods, our results confirm that strategies optimal for individual models are not necessarily optimal for the ensemble. For WRN-16-4 on CIFAR-10, applying temperature scaling to each member individually degrades the final ensemble's calibration (ECE), even though it improves the calibration of the single models (detailed in Appendix E). Conversely, the effect of joint scaling on ensemble ECE was inconsistent in our experiments. While it improved calibration for the WRN, MLP, and BiLSTM configurations, we found it could be detrimental for the GCN model, with the magnitude of this effect also varying significantly. The complete numerical data for these findings are available in Appendix E (Tables 5 through 10).

**Conclusions**   Our findings confirm that joint temperature scaling is preferable to individual scaling. This result is consistent with previous work (Wu & Gales, 2021). While joint scaling offers a path to better calibration, particularly on multi-class tasks like WRN/CIFAR-10, the improvements are often modest.

The most critical factor for practitioners is the trade-off in validation set size, as reserving too much data demonstrably harms overall model performance by shrinking the training set. While the overlapping holdout strategy provides robustness against this performance degradation, especially in data-scarce scenarios, the most effective approach in our experiments was typically to use a smaller validation set (1–5%) with a standard shared holdout.

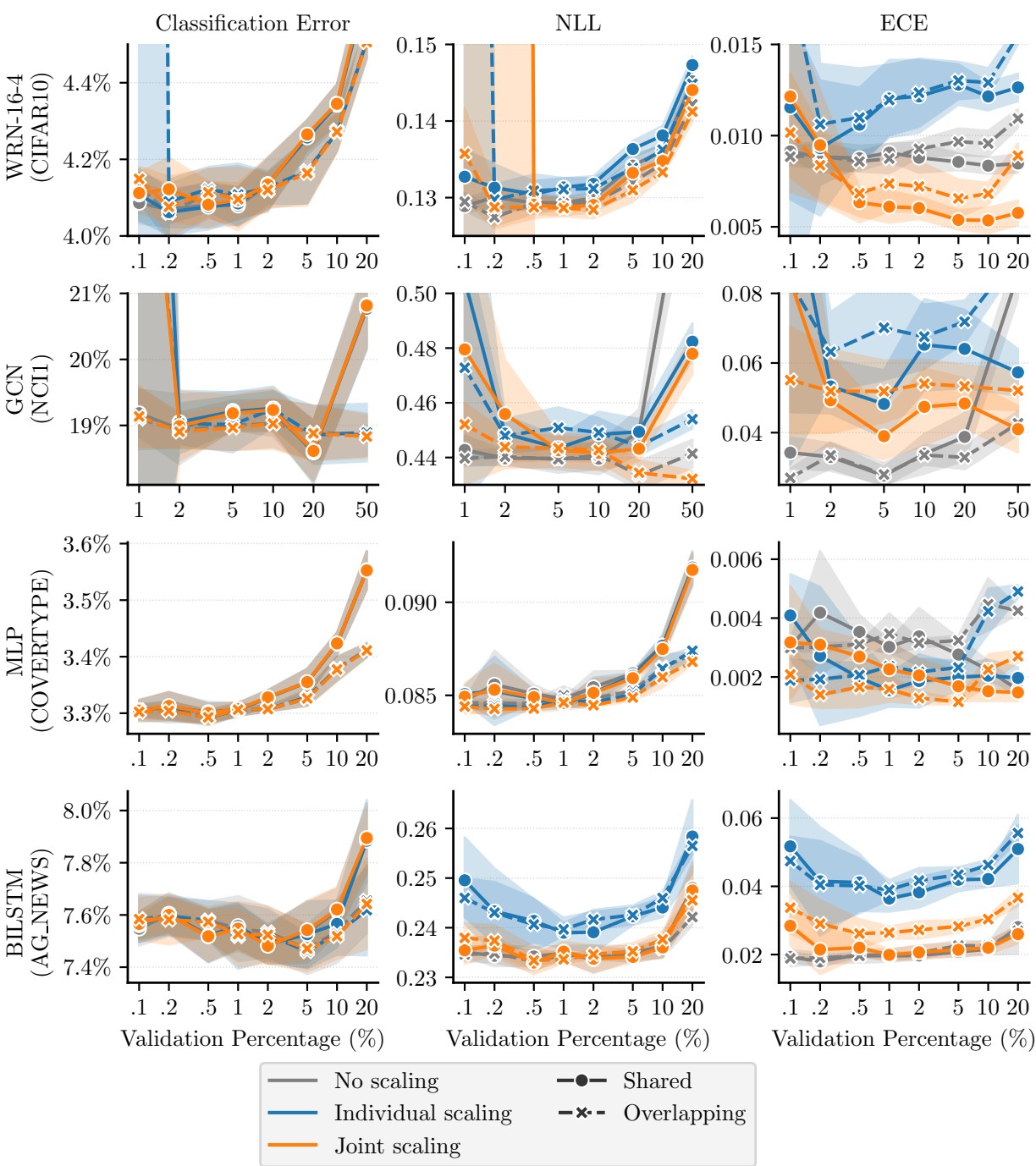

Figure 2: Temperature scaling test results for the full ensemble ($M = 4$ for WRN and GCN; $M = 8$ for MLP and BiLSTM). This plot compares different scaling approaches across varying validation percentages and holdout strategies. (WRN: Wide ResNet; GCN: Graph Convolutional Network; MLP: Multi-Layer Perceptron; BiLSTM: Bidirectional Long Short-Term Memory; NLL: negative log-likelihood; ECE: expected calibration error).

### 4.3 Early Stopping

**Purpose**  This experiment aims to compare the effectiveness of individual versus joint early stopping strategies for deep ensembles. We assess their impact on key metrics including NLL, classification error, ECE, training duration, and ensemble diversity, utilizing different validation holdout methods and varying validation set sizes.

**Method**  We compared individual and joint early stopping based on validation NLL using the Adam optimizer without weight decay and a patience of 10 epochs. Shared, disjoint, and overlapping holdout strategies were compared across various validation set percentages. To allow fair comparison of training duration across experiments with different validation set sizes, we report stopping times in step-normalized epochs (experimental parameters are detailed in Appendix A.3).

**Results**  The results in Figure 3 show that joint early stopping consistently outperforms individual stopping, leading to lower classification error and NLL for the final ensemble across all holdout strategies. As shown in Figure 4, this performance gain is attributed to a longer training duration. The impact on ensemble calibration (ECE) is also largely positive, with improvements for the WRN, MLP, and BiLSTM models, while the GCN model's ECE is maintained.

This extended training highlights the ensemble optimality gap: while the longer training under joint stopping also improves the classification error of the average individual model, it generally comes at the cost of worse calibration and higher NLL at the individual level (see Appendix F). The effect of longer training on ensemble diversity is inconsistent across our experiments; it is lower for the MLP, higher for the BiLSTM, and largely unchanged for the WRN and GCN.

Our experiments also underscore the trade-off in validation set size, where using too much data harms performance by reducing the training set, and using too little can lead to premature stopping. Here, the choice of holdout strategy is important. The disjoint holdout strategy, which prevents joint evaluation, consistently performs the worst. In contrast, for the GCN and MLP models at high validation percentages, the overlapping holdout strategy with joint stopping outperforms the shared holdout. This suggests its superior data utilization and a potential increase in ensemble diversity can mitigate the performance loss from a large validation set.

**Conclusions**  Monitoring the entire ensemble's performance on a validation set to guide early stopping provides a clear advantage over stopping members individually. This strategy allows individual members to train beyond their individual optima, benefiting collective ensemble generalization and demonstrating a clear ensemble optimality gap. While this can introduce an accuracy-calibration trade-off for the individual members, the final ensemble's performance is consistently improved.

From a practical standpoint, we recommend joint early stopping with a shared holdout set, as it is not significantly more complex to implement than individual stopping when models are trained on the same device. However, in data-scarce settings that necessitate a large validation percentage, the overlapping holdout strategy presents a valuable alternative that is worth exploring. We also find that applying post-hoc temperature scaling provides no significant additive benefit to ensembles already regularized with joint early stopping, as detailed in Appendix G.

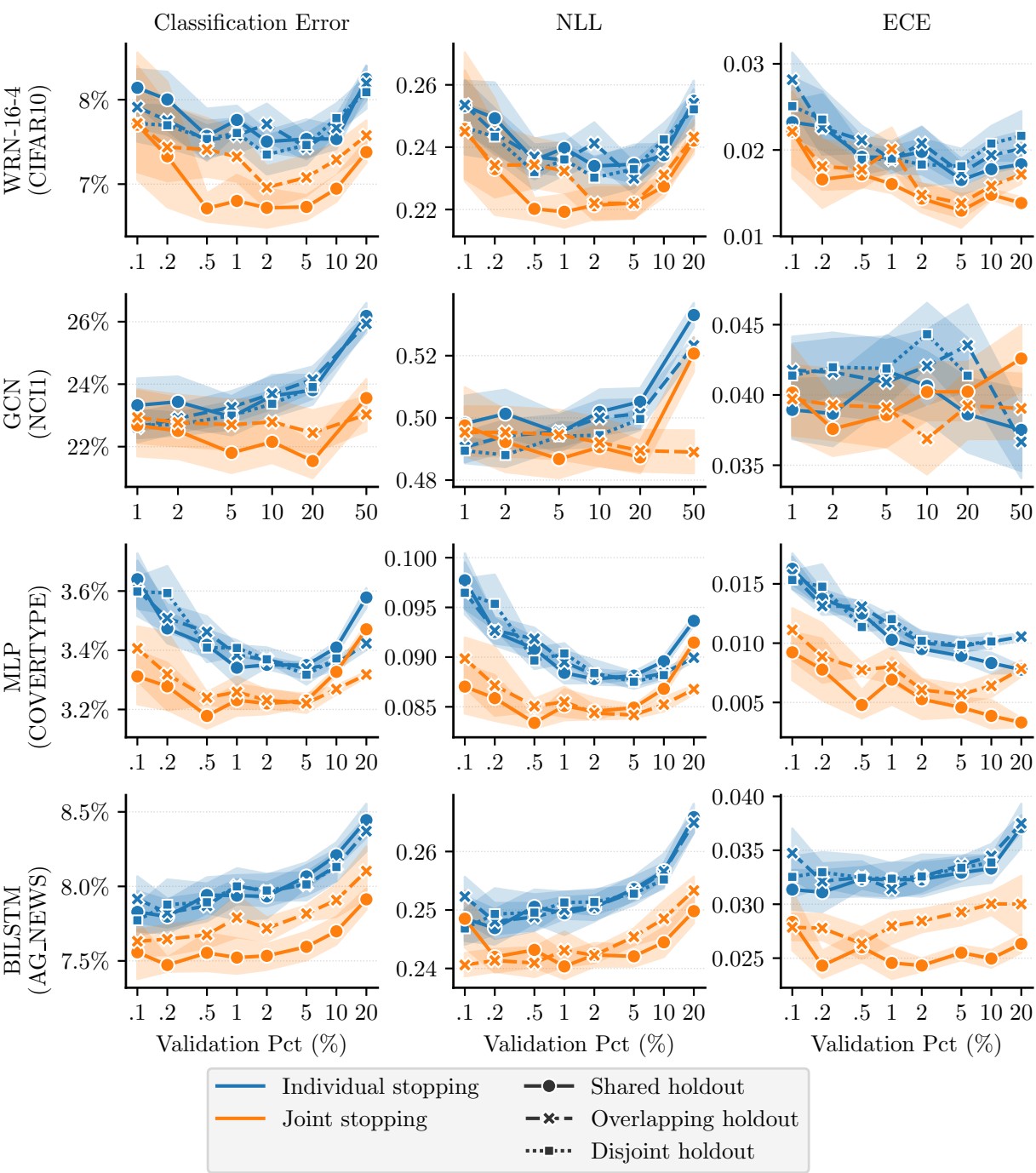

Figure 3: Early stopping test performance for the full ensemble ($M = 4$ for WRN and GCN; $M = 8$ for MLP and BiLSTM), comparing different early stopping strategies across all holdout types. (WRN: Wide ResNet; GCN: Graph Convolutional Network; MLP: Multi-Layer Perceptron; BiLSTM: Bidirectional Long Short-Term Memory; NLL: negative log-likelihood; ECE: expected calibration error).

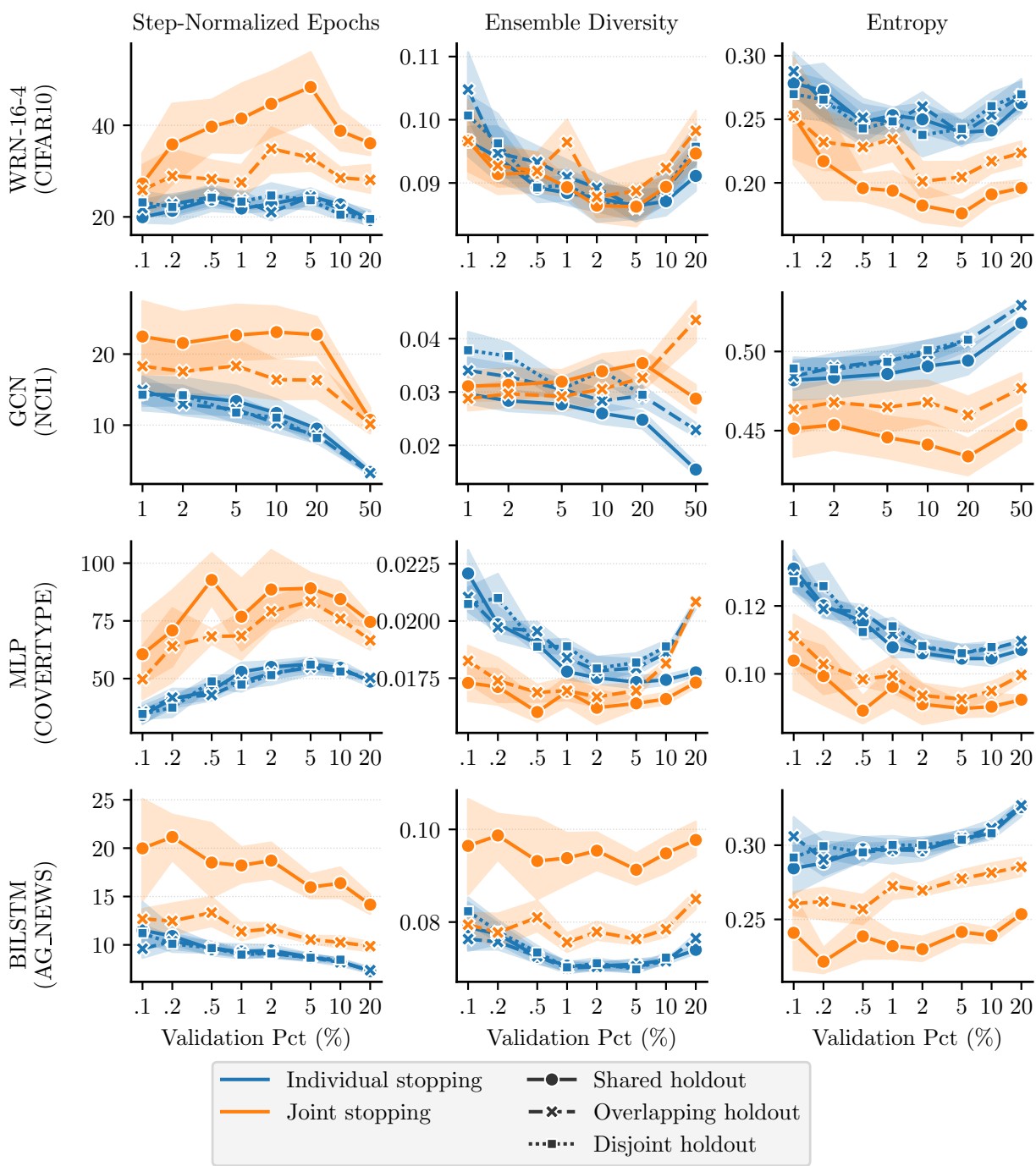

Figure 4: Additional insights into the early stopping strategies for the full ensemble. We show the stopping epoch (normalized by training steps), alongside the resulting test set ensemble diversity and predictive entropy.(WRN: Wide ResNet; GCN: Graph Convolutional Network; MLP: Multi-Layer Perceptron; BiLSTM: Bidirectional Long Short-Term Memory).

### 4.4 Validation Strategies and Data Leakage in BatchEnsembles

**Purpose**  While standard deep ensembles consist of independent models, parameter-efficient methods like BatchEnsemble (Wen et al., 2020b) utilize extensive parameter sharing (see Appendix A.1.5). This structural difference raises a critical question: Can non-shared holdouts (disjoint or overlapping) be safely used, or does parameter sharing lead to data leakage, where information from one member's validation set influences another? We investigate this potential failure mode and hypothesize that the risk is linked to the initialization of the member-specific parameters, which governs the functional diversity of the ensemble.

**Method**  We implemented BatchEnsemble ($M = 4$) on CIFAR-10 using a WRN-16-4 architecture. This method employs shared *slow weights* modulated by member-specific, rank-1 *fast weights* (details in Appendix A.1.5). We investigated three fast weight initialization strategies (Gaussian $\sigma = 0.1$, $\sigma = 0.5$, and random sign $\pm 1$) across shared, disjoint, and overlapping holdout structures (2% validation split). A key difference from standard ensembles is that BatchEnsemble's shared parameters necessitate simultaneous training termination for all members. For the shared and overlapping holdouts, which permit joint evaluation, we used an early stopping criterion based on the joint ensemble's NLL. However, as the disjoint strategy lacks a common validation set, we instead stopped training based on the average of individual NLLs across their respective validation sets. Following best practices (Wen et al., 2020a), separate batch normalization layers were used for each member (experimental parameters are detailed in Appendix A.3).

**Results**  Our findings, presented in Table 2, reveal that the fast weight initialization strategy is the critical factor determining both performance and the validity of using non-shared holdouts. Among the BatchEnsemble configurations, random sign initialization consistently performs the best, achieving substantially better NLL, ECE, and much higher ensemble diversity than the Gaussian initializations. While it does not fully match the performance of the standard Deep Ensemble baseline, it represents a significant improvement over a single model.

Figure 5 explains why the Gaussian initializations are particularly unreliable with non-shared holdouts. For these configurations, the validation and training metrics for individual models closely align, while the test metrics diverge significantly. This lack of a typical generalization gap is strong evidence of data leakage, where information from one member's validation set influences others through the shared parameters, leading to poor generalization. In contrast, random sign initialization exhibits a more typical and desirable gap between training, validation, and test performance, indicating that it successfully mitigates this leakage risk.

**Conclusions**  Our investigation confirms that non-shared holdout strategies can be used with BatchEnsemble, but this is critically dependent on the initialization promoting sufficient functional diversity. Random sign initialization is strongly preferable, as it mitigates data leakage and allows the model to behave more like a standard deep ensemble. Conversely, Gaussian initializations lead to unreliable performance and poor calibration when used with non-shared validation sets.

This highlights a crucial consideration for practitioners: parameter-efficient ensembles like BatchEnsemble can have hidden complexities and may not behave identically to their standard counterparts. The choice of a seemingly minor detail, like the initialization of member-specific parameters, can determine the validity of a validation strategy. These findings underscore that more work is needed to understand the dynamics of efficient ensembles and to establish robust best practices that ensure they provide reliable improvements.

Table 2: Test set performance of BatchEnsemble ($M = 4$, WRN-16-4 on CIFAR-10) compared to a standard Deep Ensemble (DE) baseline. The table compares holdout structures and fast weight initializations for BatchEnsemble. Bold values are the best among the BatchEnsemble configurations (within 1.96 SEM). The DE baseline is not included in the comparison.

| Metric | Deep Ensemble (Shared Holdout) | $\mathcal{N}(1, 0.1)$ Shared | Overlapping | Disjoint | $\mathcal{N}(1, 0.5)$ Shared | Overlapping | Disjoint | Random sign ($\pm 1$) Shared | Overlapping | Disjoint |
|---|---|---|---|---|---|---|---|---|---|---|
| Err. (%) ↓ | 6.7 $\pm 0.1$ | 11.0 $\pm 0.6$ | 8.2 $\pm 0.2$ | 8.0 $\pm 0.1$ | 10.2 $\pm 0.3$ | **7.8** $\pm 0.1$ | **7.8** $\pm 0.1$ | **7.9** $\pm 0.1$ | 8.1 $\pm 0.2$ | 8.6 $\pm 0.2$ |
| NLL ↓ | 0.221 $\pm 0.003$ | 0.440 $\pm 0.008$ | 0.456 $\pm 0.007$ | 0.446 $\pm 0.004$ | 0.363 $\pm 0.004$ | 0.430 $\pm 0.011$ | 0.432 $\pm 0.010$ | **0.258** $\pm 0.003$ | **0.259** $\pm 0.004$ | 0.269 $\pm 0.005$ |
| ECE ↓ | 0.014 $\pm 0.001$ | 0.058 $\pm 0.002$ | 0.055 $\pm 0.001$ | 0.054 $\pm 0.001$ | 0.041 $\pm 0.002$ | 0.050 $\pm 0.001$ | 0.050 $\pm 0.001$ | **0.020** $\pm 0.001$ | **0.022** $\pm 0.001$ | 0.023 $\pm 0.001$ |
| Diversity ↑ | 0.086 $\pm 0.001$ | 0.007 $\pm 0.001$ | 0.012 $\pm 0.001$ | 0.011 $\pm 0.001$ | 0.026 $\pm 0.002$ | 0.017 $\pm 0.001$ | 0.017 $\pm 0.001$ | **0.106** $\pm 0.002$ | **0.106** $\pm 0.001$ | **0.104** $\pm 0.001$ |
| Entropy | 0.182 $\pm 0.008$ | 0.137 $\pm 0.021$ | 0.064 $\pm 0.004$ | 0.062 $\pm 0.001$ | 0.160 $\pm 0.015$ | 0.067 $\pm 0.003$ | 0.067 $\pm 0.003$ | 0.240 $\pm 0.008$ | 0.249 $\pm 0.010$ | 0.282 $\pm 0.010$ |
| Norm. Epochs | 44.69 $\pm 3.82$ | 12.45 $\pm 1.85$ | 53.12 $\pm 6.34$ | 52.43 $\pm 4.56$ | 14.21 $\pm 1.12$ | 58.02 $\pm 5.28$ | 59.19 $\pm 6.84$ | 39.69 $\pm 2.66$ | 37.04 $\pm 3.18$ | 27.34 $\pm 1.86$ |

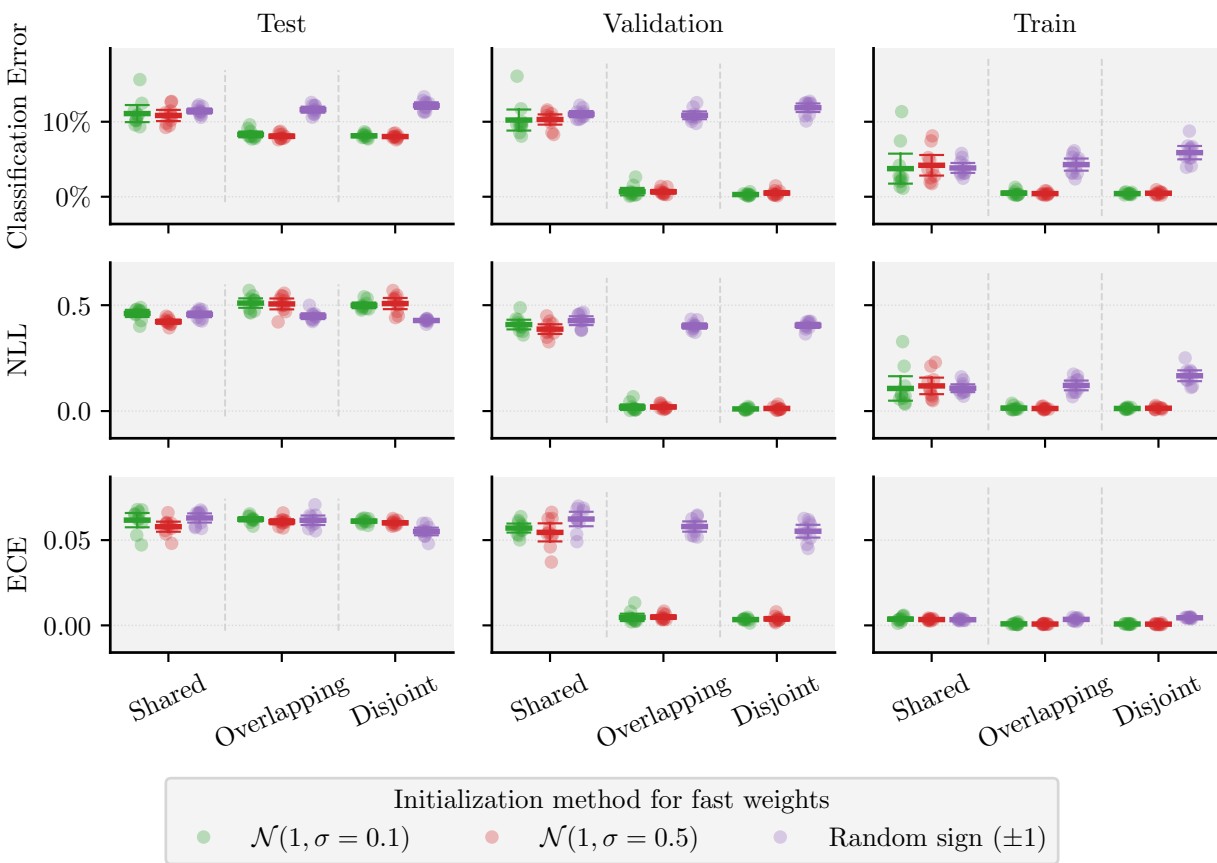

Figure 5: Average individual model performance within BatchEnsemble ($M = 4$, WRN-16-4 on CIFAR-10), comparing different initialization strategies for *fast weights* across shared, overlapping, and disjoint holdout strategies. The results for classification error, NLL, and ECE are shown for the test, validation, and training sets. Notably, for Gaussian initialization with overlapping and disjoint holdouts, the close alignment of validation and training performance (as opposed to test performance) suggests potential data leakage between ensemble members. (WRN: Wide ResNet; NLL: negative log-likelihood; ECE: expected calibration error).

## 5 Discussion and Conclusion

This study investigated the practical impact of adopting an ensemble-aware perspective when tuning hyperparameters and applying calibration or regularization techniques, specifically focusing on the potential mismatch between individually optimal settings—the *ensemble optimality gap*. Our experiments across weight decay tuning, temperature scaling, early stopping, different validation strategies, and BatchEnsemble provide several insights into optimizing deep ensemble performance.

Our findings regarding weight decay tuning confirm that the optimal value for a single model provides a strong baseline. However, for most of the benchmarks, joint tuning revealed that a slightly lower level of regularization was beneficial for the ensemble, though the specific balance between improving classification error and calibration (ECE) was dependent on the model and dataset. While jointly tuning the entire ensemble for weight decay is computationally expensive, our results suggest that practitioners can consider the single-model optimum as a good starting point and explore slightly lower values.

For post-hoc calibration via temperature scaling, our findings show that a joint optimization strategy is preferable to calibrating members individually. This is consistent with prior work (Rahaman & Thiéry, 2021; Wu & Gales, 2021). While applying a single temperature to the joint ensemble generally improved calibration over an uncalibrated baseline, the magnitude of this improvement was often modest and task-dependent. Critically, these experiments highlight the need for a sufficiently large validation set for robust temperature estimation; however, using significantly more data than necessary yielded little further improvement in calibration while negatively impacting overall model performance due to the reduced training set size.

Perhaps the clearest advantage for joint optimization was observed with early stopping. Monitoring the performance of the entire ensemble and stopping only when its collective performance ceased to improve consistently led to better ensemble NLL and classification error compared to stopping members individually based on their own optima. This aligns with early ensemble theory (Sollich & Krogh, 1995), suggesting that allowing individual members to train longer (potentially slightly past their individual optimal stopping points) benefits the ensemble's generalization capacity. Although such extended training can negatively impact the calibration of individual members considered in isolation, the final ensemble calibration was often improved (e.g., WRN-16-4, MLP, BiLSTM) or maintained (e.g., GCN) compared to using individual stopping.

Conversely, tuning on individual models offers significant practical benefits that explain its widespread use. The approach is simple, requires no changes to standard single-model training pipelines, and allows for modular development where members can be trained and debugged independently. Adopting a joint optimization strategy, in contrast, introduces complexities. It can increase implementation overhead, require more sophisticated resource management to handle multiple models simultaneously during validation, and necessitates a carefully chosen validation strategy. Despite these added hurdles, our findings show that the performance gains from joint optimization can be compelling. For computationally inexpensive procedures like early stopping, in particular, the trade-off is often favorable, as the modest increase in implementation complexity can unlock clear improvements in generalization.

The choice of validation data strategy critically determines the feasibility of joint optimization techniques and involves fundamental trade-offs. A shared holdout enables straightforward joint evaluation of the full ensembles but requires reserving a common validation set that is never used for training by any member. Conversely, the disjoint holdout strategy maximizes the utilization of the available non-test data pool for training across the ensemble-ensuring every data point contributes to training all but one member-but completely precludes joint evaluation on validation data unseen by all members being evaluated. Our results for early stopping and temperature scaling suggest that the benefits derived from enabling robust joint evaluation outweigh the potential advantages of maximizing this training data utilization. The proposed overlapping holdout strategy offers a middle ground: similar to disjoint, it ensures all non-test data is used for training somewhere within the ensemble, but by creating specific overlaps within the validation sets assigned to each member, it permits pairwise joint evaluation. This makes it a practical compromise, particularly in low-data regimes where reserving a fully shared holdout might be too costly because it prevents the entire portion of data from being used in training by any model.

The importance of considering ensemble interactions also extends to parameter-efficient methods like BatchEnsemble. Our investigation underscores the paramount importance of the initialization strategy for the member-specific *fast weights* (the trainable rank-1 vectors), a detail potentially overlooked. We found that random sign initialization (assigning $\pm 1$ values) proved crucial for achieving high ensemble diversity and good calibration, behaving much like a standard ensemble. This aligns with some earlier observations regarding its effectiveness (Wenzel et al., 2020). In stark contrast, Gaussian initialization (i.e., sampling from $\mathcal{N}(1, \sigma^2)$), although used in some large-scale applications (Tran et al., 2022; Dehghani et al., 2023), performed poorly in our experiments, showing lower diversity and signs of data leakage with non-shared holdouts. The potential confusion regarding initialization practices is highlighted by reports using ambiguous descriptions like "random sign initialization... of -0.5" (meaning Gaussian initialization) at large scale (Dehghani et al., 2023). Given the stark difference we observed, and the lack of direct comparisons at scale in cited works, further investigation into the impact of initialization on BatchEnsemble's effectiveness seems warranted. Our findings strongly suggest that proper (random sign) initialization leads to robust BatchEnsemble behavior, crucially mitigating data leakage issues with non-shared holdouts and thereby enabling the effective and reliable use of different validation strategies, including meaningful joint evaluation when the structure allows (like overlapping holdouts).

Across these diverse experiments, a unifying principle emerges: the significance of the *ensemble optimality gap* and the practical value of adopting an ensemble-aware perspective. Evaluating and optimizing based on the joint ensemble's behavior during validation-dependent procedures consistently led to ensembles that were more accurate, better calibrated, or both. Our results suggest practitioners should prioritize robust joint evaluation strategies-especially for computationally inexpensive procedures like early stopping and temperature scaling-rather than solely relying on individually tuned components or potentially complex methods aimed at explicitly maximizing diversity during training, which may not always be necessary or beneficial, particularly for large models (Abe et al., 2022; 2024). Ultimately, the ensemble is the final predictor, and evaluating it directly during optimization yields better results.

Based on these findings, we offer the following practical recommendations for practitioners training deep ensembles:

- **Weight Decay:** Start by finding the optimal weight decay for a single model, as this provides a strong baseline. To account for the ensemble optimality gap, consider also evaluating the ensemble performance with nearby weight decay values (particularly slightly lower ones), as the optimal setting for the ensemble may differ depending on the model, dataset, and target metric.

- **Temperature Scaling:** Always optimize and apply temperature scaling *jointly* based on the ensemble's performance (e.g., minimizing NLL) on a validation set. Avoid calibrating members individually. Crucially, use only a reasonably small validation set; dedicating too much data can unnecessarily harm overall model performance.

- **Early Stopping:** Monitor the validation performance of the *entire ensemble* to determine the stopping point. This allows the ensemble to train longer and achieve better performance compared to stopping based on individual member optima. Ensure the validation set size is appropriate.

- **Validation Strategy:** Use a *shared holdout* set whenever possible to allow for direct joint evaluation. If data is extremely limited, making a shared holdout prohibitively costly, the *overlapping holdout* strategy offers a viable alternative that retains some joint evaluation capability while maximizing data use.

- **BatchEnsemble:** Strongly prefer *random sign initialization* for the fast weights over Gaussian initialization to maximize ensemble diversity, calibration, and robustness, particularly when using disjoint or overlapping holdout strategies.

While our findings show consistent trends across four diverse benchmarks (image, graph, tabular, and text) and multiple ensemble sizes ($M = 4$ and $M = 8$), this study has limitations that open avenues for future work. We did not investigate our hypotheses on very large-scale models and datasets, where the computational

cost of joint tuning might be prohibitive and ensembling dynamics could differ. Similarly, whether Gaussian initialization for BatchEnsemble could yield sufficient diversity in such large models remains an open question, given the lack of direct comparisons in the literature we surveyed. The implications of our findings for other types of ensembles (e.g., BNNs, MC-dropout) also warrant investigation. Future work could explore these joint optimization dynamics in other domains, develop more cost-effective heuristics for joint tuning, investigate variations of the overlapping holdout strategy—such as structures enabling higher-order joint evaluation (e.g., among member triplets)—and further study the interplay between model scale and the optimal validation strategy.

In conclusion, this work highlights the practical importance of considering ensemble effects during the tuning and calibration process. By adopting an ensemble-aware perspective and leveraging joint evaluation, particularly for computationally efficient techniques like early stopping and temperature scaling, practitioners can build more accurate and reliable deep ensemble models.

### Acknowledgments

The authors acknowledge support from the Novo Nordisk Foundation under grant no NNF22OC0076658 (Bayesian neural networks for molecular discovery).

We acknowledge the Danish e-infrastructure Consortium (DeiC) for awarding this project access to the LUMI supercomputer, owned by the EuroHPC Joint Undertaking, hosted by CSC (Finland) and the LUMI consortium.

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

# A  Experimental Setup Details

The code to reproduce the experiments presented in this paper is publicly available at `https://github.com/lauritsf/ensemble-optimality-gap`. All experiments were conducted on the LUMI supercomputer, where each ensemble was trained on a single Graphics Compute Die (GCD) on a LUMI-G node, which is equipped with AMD MI250x GPUs.

## A.1  Model Details

### A.1.1  Wide ResNet-16-4 (WRN-16-4)

For our experiments on the CIFAR-10 dataset, we utilize the WRN-16-4 architecture (Zagoruyko & Komodakis, 2016) as one of our base models for deep ensembles. WRN architectures are characterized by their wider convolutional layers and reduced depth compared to traditional ResNet architectures. Specifically, WRN-16-4 denotes a WRN with 16 convolutional layers and a widening factor of 4. In our implementation, we use the WRN variant where the placement of batch normalization, ReLU activation, and convolution layers follows the order: batch normalization - ReLU - convolution. We do not employ dropout regularization in our models.

### A.1.2  Graph Convolutional Network (GCN)

For our experiments on the NCI1 dataset, we utilize a four-layer GCN, based on the architecture presented in Sui et al. (2022), but extended to four GCN layers. The network consists of an initial feature transformation layer, followed by four GCN layers, and finally two fully connected layers.

We use ReLU activations after the initial feature transformation and each of the four GCN layers. Batch normalization is applied to the input features before the initial transformation, after each of the four GCN layers, and before each of the two fully-connected layers.

Global sum pooling is applied after the final GCN layer to obtain a graph-level representation. This representation is then passed through a sequence of two fully-connected layers. The first fully connected layer applies batch normalization, followed by a ReLU activation and a linear transformation. The second fully-connected layer is the classification layer and produces the final output logits.

### A.1.3  Multilayer Perceptron (MLP)

For the Covertype dataset, we use an MLP with three hidden layers, each with 1024 units. The architecture for each hidden block consists of a linear layer without bias, followed by batch normalization and a ReLU activation. The output layer is a linear transformation to the 7 classes of the dataset.

### A.1.4  Bidirectional LSTM (BiLSTM)

For the AG News classification task, we use a BiLSTM network. The model's architecture consists of three main components: an initial embedding layer that converts tokens into 128-dimensional vectors, a single BiLSTM layer with a hidden size of 256 units for each direction (resulting in a 512-dimensional output vector for each token), and a self-attention mechanism that computes a weighted average over the BiLSTM's output sequence to create a single context vector.

This context vector is then passed to a final linear layer that maps it to the 4 output classes. The input text is processed using the pretrained GPT-2 tokenizer.

### A.1.5  BatchEnsemble Implementation

BatchEnsemble (Wen et al., 2020b) modifies a base network to efficiently train an ensemble using shared weights $W$ (slow weights). For each shared weight $W$, member-specific weights $W_i$ ($i = 1...M$) are generated using trainable rank-1 vectors $\mathbf{r}_i$ and $\mathbf{s}_i$ (fast weights) as $W_i = W \circ (\mathbf{r}_i \mathbf{s}_i^T)$, where $\circ$ denotes the Hadamard

(element-wise) product. Crucially, this rank-1 structure allows the forward pass computations for all $M$ ensemble members to be efficiently vectorized into a single operation, enabling parallel execution on hardware accelerators (Wen et al., 2020b). The initialization of these fast weights is known to be important for performance. As investigated in Section 4.4, our experiments specifically compare three initialization strategies for $\mathbf{r}_i$ and $\mathbf{s}_i$: initializing elements from a Gaussian distribution with mean 1 and standard deviation $\sigma$ (either $\sigma = 0.1$ or $\sigma = 0.5$), versus initializing elements randomly as $\pm 1$ (random sign / Rademacher distribution). Consistent with recommendations for achieving good performance with BatchEnsemble (Wen et al., 2020a), we use separate batch normalization layers for each ensemble member throughout our experiments.

## A.2 Dataset Details

### A.2.1 CIFAR-10

We conduct experiments on the CIFAR-10 dataset (Krizhevsky, 2009), a widely used benchmark for image classification. CIFAR-10 consists of 60,000 32x32 color images in 10 classes, with 6,000 images per class. There are 50,000 training images and 10,000 test images, and we use the original test set for final performance evaluation in all experiments.

Following common practice for CIFAR-10, we apply the following preprocessing steps: random cropping to size 32 with a padding of 4 pixels, random horizontal flipping with a probability of 0.5, and normalization. The normalization constants (mean and standard deviation for each color channel) are calculated based on the actual training split to prevent contamination from the test and validation data.

### A.2.2 NCI1

We conduct experiments on the NCI1 dataset (Shervashidze et al., 2011; Wale et al., 2008), a graph classification dataset with two classes. There are 4110 graphs. We randomly split the dataset into a training set (80%) and a test set (20%) using stratified sampling to ensure class balance, resulting in 3288 training graphs and 822 test graphs. The test set is kept fixed across all experiments. We do not apply any specific preprocessing to the node features or graph structure beyond what is inherent in the dataset.

### A.2.3 Covertype

We use the Covertype dataset (Blackard & Dean, 1999) from the UCI repository, a large-scale tabular benchmark for predicting forest cover type from cartographic variables. The dataset contains 581,012 samples, which we split into 464,809 for training and 116,203 for testing. It includes 54 features and 7 classes. We perform input normalization on the features, using the mean and standard deviation calculated from each model's respective training data.

### A.2.4 AG News

Our text classification experiments use the AG News dataset (Zhang et al., 2015), which consists of 120,000 training samples and 7,600 test samples for classifying news articles into 4 balanced classes. The input text is processed using the pretrained GPT-2 tokenizer (Radford et al., 2019) with a max length corresponding to the longest sequence in the dataset.

## A.3 Experimental Setup Details

Across all experiments, the batch size was set to 128. The ensemble size was $M = 4$ for the WRN-16-4 and GCN models, and $M = 8$ for the MLP and BiLSTM models.

**Hyperparameter Tuning (Weight Decay)** For the weight decay tuning experiments, models were trained using SGD with a momentum of 0.9 and a cosine annealing learning rate schedule. We performed a grid search over a range of log-spaced weight decay values, always including 0. We used 100 epochs (200 for GCN), an initial learning rate of 0.1 (0.3 for MLP), and 5 random seeds (20 for GCN).

**Temperature Scaling** For post-hoc temperature scaling, we used base models trained with the optimal single-model weight decay. The temperature parameter was optimized using the L-BFGS algorithm with a maximum of 100 iterations. We used 10 seeds (5 for MLP and BiLSTM) and a validation percentage range of 0.1–20% (1–50% for GCN).

**Early Stopping** The early stopping experiments were conducted using the Adam optimizer with a patience of 10 epochs and no weight decay, using the validation NLL as the stopping criterion. The learning rate was $1 \times 10^{-3}$ ($1 \times 10^{-4}$ for BiLSTM) and experiments were run on 10 seeds (50 for GCN). The range of validation percentages was 0.1–20% (1–50% for GCN).

**BatchEnsemble** The BatchEnsemble experiment was performed with a WRN-16-4 on CIFAR-10 with an ensemble size of $M = 4$. We used the Adam optimizer with a learning rate of $1 \times 10^{-3}$, no weight decay, and an early stopping patience of 10 epochs on a 2% validation split. The experiment was run across 10 seeds. We tested three fast weight initializations: Gaussian ($\mathcal{N}(\mu = 1, \sigma = 0.1)$ and $\mathcal{N}(\mu = 1, \sigma = 0.5)$) and random sign ($\pm 1$). Consistent with best practices, a separate batch normalization layer was used for each ensemble member.

## B Diversity Metric

The ensemble diversity metric used in this paper (Sections 2.1, 4.3, 4.4) quantifies the disagreement among ensemble members for a given input data point $i$. It is defined as the difference between the entropy of the average predicted probability vector $\bar{\boldsymbol{p}}_i = \frac{1}{M} \sum_{m=1}^{M} \boldsymbol{p}_i^m$ and the average entropy of the individual member predictions $\boldsymbol{p}_i^m$:

$$D_i = H(\bar{\boldsymbol{p}}_i) - \frac{1}{M} \sum_{m=1}^{M} H(\boldsymbol{p}_i^m)$$

where $H(\boldsymbol{p}) = -\sum_c \boldsymbol{p}[c] \log \boldsymbol{p}[c]$ is the Shannon entropy.

This measure is directly related to the Kullback-Leibler (KL) divergence. It can be shown that the diversity defined above is equal to the average KL divergence from the individual member predictions $\boldsymbol{p}_i^m$ to the mean ensemble prediction $\bar{\boldsymbol{p}}_i$:

$$D_i = \frac{1}{M} \sum_{m=1}^{M} D_{KL}(\boldsymbol{p}_i^m || \bar{\boldsymbol{p}}_i)$$

where $D_{\mathrm{KL}}(\boldsymbol{p} || \bar{\boldsymbol{p}}) = \sum_c \boldsymbol{p}[c] \log(\boldsymbol{p}[c] / \bar{\boldsymbol{p}}[c])$. Therefore, this diversity metric intuitively quantifies how much, on average, the probability distribution predicted by an individual member diverges from the consensus prediction of the ensemble. Higher values indicate greater disagreement among ensemble members.

## C Supplementary Results for Weight Decay Tuning

This section supplements the weight decay tuning experiments (Section 4.1) with a detailed breakdown of the average test performance for the individual models that constitute the ensembles. Table 3 presents the classification error, NLL, and ECE for these individual models, comparing the two optimized weight decay strategies: using the value optimized for a single model's performance (Single-Model Opt.) versus the value optimized for the final ensemble's performance (Ensemble Opt.).

For completeness, Table 4 details the performance when no weight decay is applied. The results directly compare an average single model against the full ensemble, highlighting that the ensemble is significantly more robust to the complete lack of regularization, where it consistently outperforms its individual components.

Table 3: Average test performance of the individual models that constitute the ensembles. Optimal weight decay (WD*) values were determined by minimizing validation NLL for either a single model or the full ensemble. For each metric, values in bold are not statistically different from the best result (within 1.96 standard errors of the mean).

| Model / Dataset | WD* | Metric | Single-Model Opt. Single Model | Ensemble Opt. Single Model |
|---|---|---|---|---|
| WRN-16-4 CIFAR10 | S: 7.50e-04 E: 5.62e-04 | Err. (%) ↓ NLL ↓ ECE ↓ | **4.91** $_{\pm 0.06}$ 0.160 $_{\pm 0.002}$ **0.012** $_{\pm 0.001}$ | **4.88** $_{\pm 0.07}$ **0.157** $_{\pm 0.002}$ 0.013 $_{\pm 0.001}$ |
| GCN NCI1 | S: 3.16e-03 E: 2.74e-03 | Err. (%) ↓ NLL ↓ ECE ↓ | 20.10 $_{\pm 0.17}$ **0.480** $_{\pm 0.004}$ **0.042** $_{\pm 0.002}$ | **19.28** $_{\pm 0.20}$ **0.485** $_{\pm 0.006}$ 0.051 $_{\pm 0.002}$ |
| MLP COVERTYPE | S: 5.62e-05 E: 1.78e-05 | Err. (%) ↓ NLL ↓ ECE ↓ | 3.39 $_{\pm 0.02}$ **0.088** $_{\pm 0.000}$ **0.003** $_{\pm 0.000}$ | **3.27** $_{\pm 0.01}$ **0.088** $_{\pm 0.000}$ 0.006 $_{\pm 0.000}$ |
| BILSTM AG_NEWS | S: 5.62e-05 E: 5.62e-05 | Err. (%) ↓ NLL ↓ ECE ↓ | **9.96** $_{\pm 0.09}$ **0.341** $_{\pm 0.004}$ **0.045** $_{\pm 0.002}$ | **9.96** $_{\pm 0.09}$ **0.341** $_{\pm 0.004}$ **0.045** $_{\pm 0.002}$ |

Table 4: Test performance comparison between an average single model and the full ensemble when no weight decay ($WD = 0$) is used.

| Model / Dataset | Metric | No Weight Decay Avg. Single Model | Full Ensemble |
|---|---|---|---|
| WRN-16-4 CIFAR10 | Err. (%) ↓ NLL ↓ ECE ↓ | 8.08 $_{\pm 0.06}$ 0.447 $_{\pm 0.009}$ 0.058 $_{\pm 0.001}$ | 6.64 $_{\pm 0.04}$ 0.247 $_{\pm 0.004}$ 0.018 $_{\pm 0.000}$ |
| GCN NCI1 | Err. (%) ↓ NLL ↓ ECE ↓ | 19.02 $_{\pm 0.18}$ 1.072 $_{\pm 0.017}$ 0.143 $_{\pm 0.002}$ | 18.12 $_{\pm 0.15}$ 0.741 $_{\pm 0.008}$ 0.105 $_{\pm 0.001}$ |
| MLP COVERTYPE | Err. (%) ↓ NLL ↓ ECE ↓ | 3.25 $_{\pm 0.02}$ 0.092 $_{\pm 0.000}$ 0.007 $_{\pm 0.000}$ | 3.11 $_{\pm 0.01}$ 0.082 $_{\pm 0.000}$ 0.004 $_{\pm 0.000}$ |
| BILSTM AG_NEWS | Err. (%) ↓ NLL ↓ ECE ↓ | 12.09 $_{\pm 0.09}$ 1.085 $_{\pm 0.009}$ 0.103 $_{\pm 0.001}$ | 7.77 $_{\pm 0.06}$ 0.328 $_{\pm 0.003}$ 0.029 $_{\pm 0.001}$ |

## D   Comparison of Joint Temperature Scaling Methods

To address the alternative joint calibration method proposed by Rahaman & Thiéry (2021), this section provides a detailed comparison of the two primary joint temperature scaling strategies. The experiment was run using a 5% shared holdout set for validation based on a single seed (0), but otherwise follows the setting of the main temperature scaling experiments.

In the terminology of Rahaman & Thiéry (2021), our main approach, "Joint (on model logits)," corresponds to their method (C), where models are simultaneously aggregated and calibrated. The alternative, "Pool-Then-Calibrate" (method D), involves first aggregating the model outputs and then calibrating the pooled estimate. Using our notation, the Pool-Then-Calibrate prediction is given by:

$$\bar{\boldsymbol{p}}^{\text{pool}-\text{then}-\text{calibrate}}(\boldsymbol{x}) = \text{softmax}\left(\frac{\log \bar{\boldsymbol{p}}(\boldsymbol{x})}{T_{\text{pool}}}\right), \quad \text{where} \quad \bar{\boldsymbol{p}}(\boldsymbol{x}) = \frac{1}{M}\sum_{m=1}^{M}\text{softmax}(\boldsymbol{z}_m(\boldsymbol{x})) \qquad (13)$$

The results of our comparison are presented in Figure 6.

## E   Supplementary Results for Temperature Scaling

To supplement the main results in Section 4.2, this section provides a view of the temperature scaling performance. While individual scaling can improve the calibration of the individual models (e.g., for WRN-16-4), as shown in Figure 7, this benefit does not transfer to the final ensemble, whose performance is shown in the main paper in Figure 2.

Tables 5 through 10 provide the numerical results for both the ensemble and the average of individual models across all metrics.

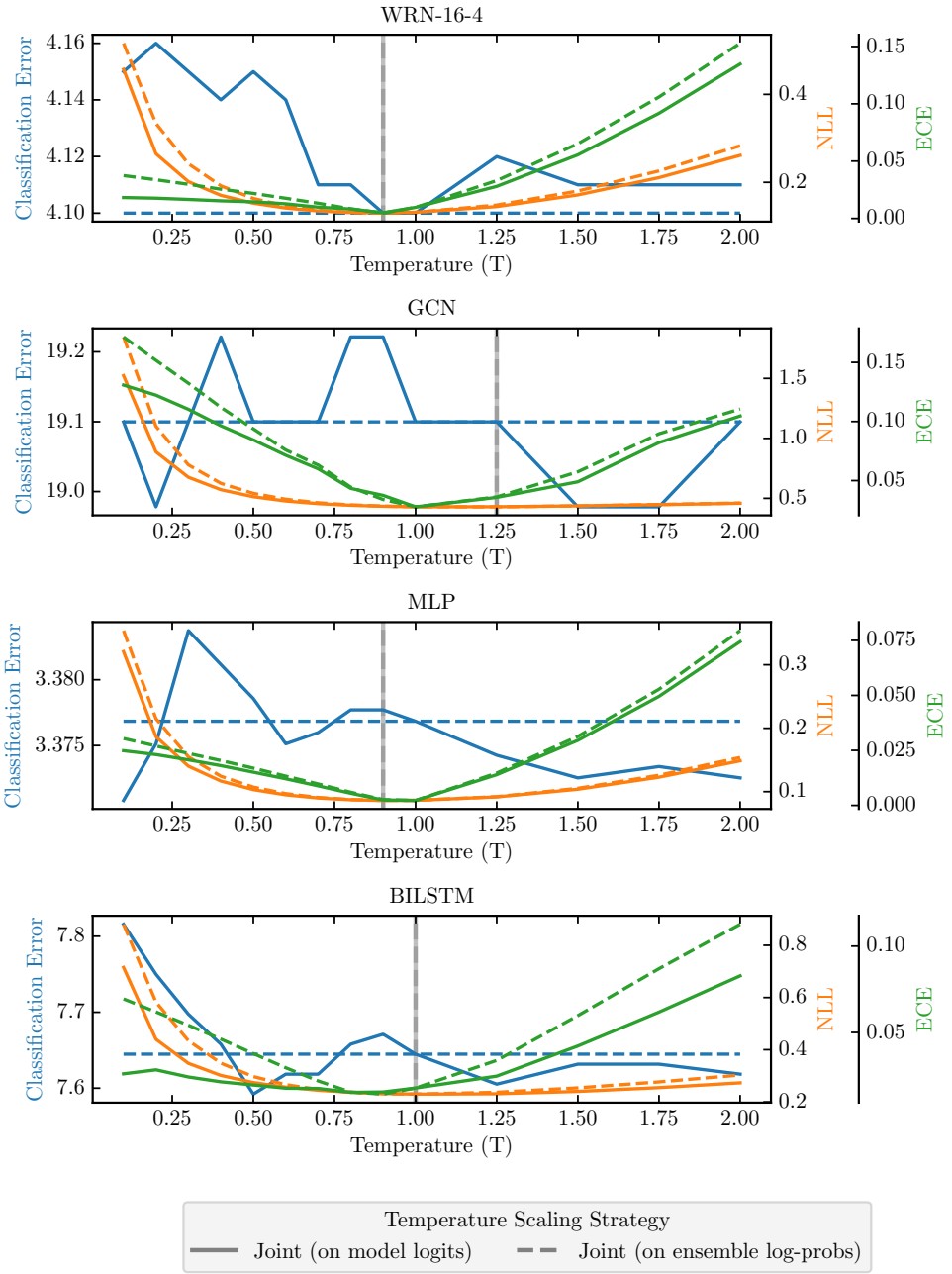

Figure 6: A comparison of two joint temperature scaling strategies. The solid line represents scaling model logits before averaging (method C in Rahaman & Thiéry (2021)), while the dashed line represents scaling the ensemble log-probabilities after averaging (method D). The vertical lines indicate the temperature that minimizes NLL for each strategy, which are nearly identical. Scaling the model logits can slightly alter the classification error, whereas scaling the log-probabilities preserves it. Despite this, the overall performance on NLL and ECE is nearly identical for both methods.

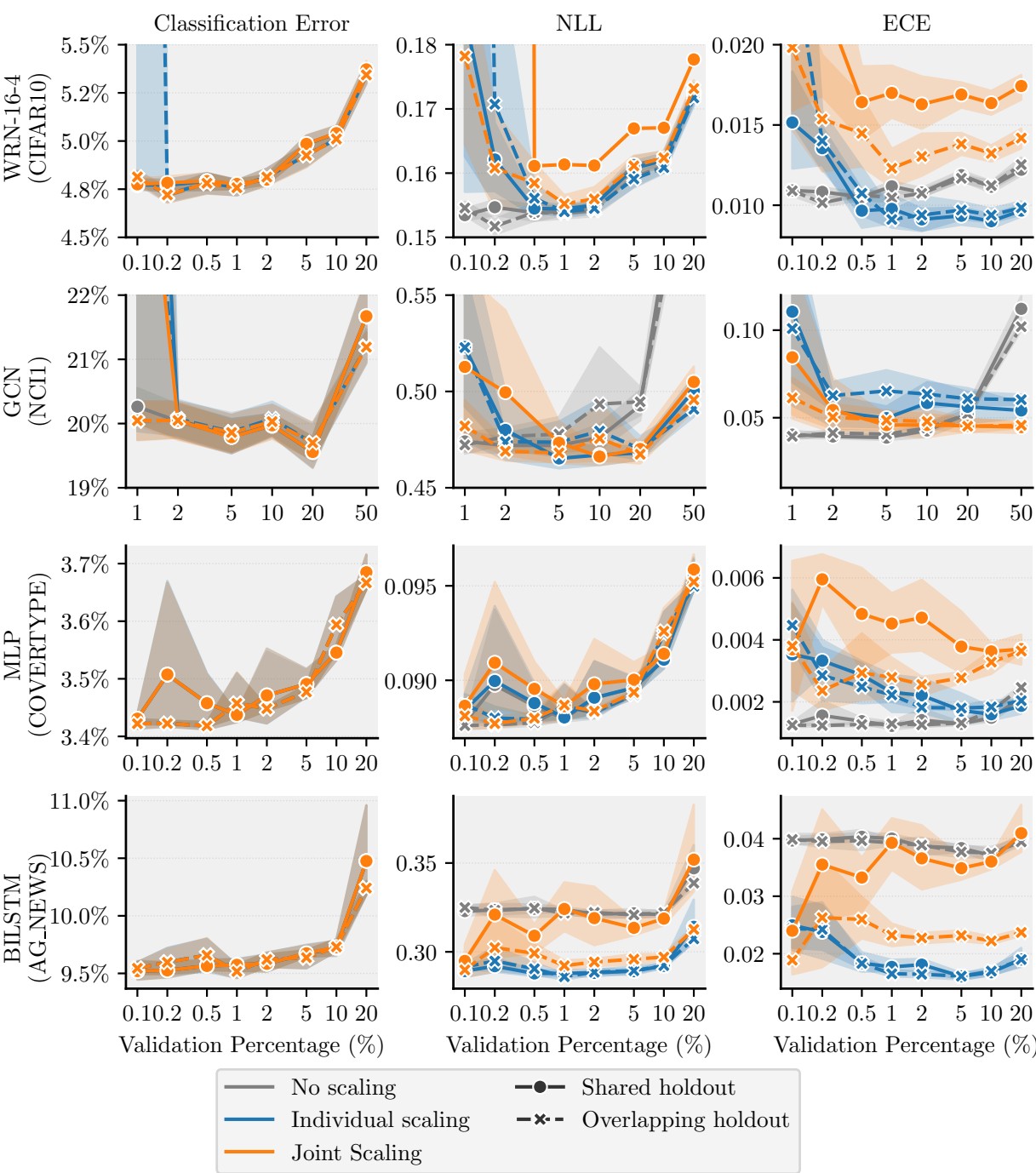

Figure 7: Test performance for the average of individual models within the ensemble, comparing different temperature scaling approaches. This figure shows results corresponding to the ensemble performance presented in Figure 2. (WRN: Wide ResNet; GCN: Graph Convolutional Network; MLP: Multi-Layer Perceptron; BiLSTM: Bidirectional Long Short-Term Memory; NLL: negative log-likelihood; ECE: expected calibration error).

Table 5: Ensemble temperature scaling results for Classification Error (%). Bold values are not statistically different from the best in each row (within 1.96 SEM).

| Model / Dataset | Val. % | Ensemble: Classification Error (%) ↓ | | | | | |
| --- | --- | --- | --- | --- | --- | --- | --- |
| | | No scaling | | Individual scaling | | Joint scaling | |
| | | **Shared** | **Overlapping** | **Shared** | **Overlapping** | **Shared** | **Overlapping** |
| | 0.1% | **4.09** ±0.03 | **4.13** ±0.02 | **4.11** ±0.03 | 13.74 ±9.58 | **4.11** ±0.03 | 4.15 ±0.02 |
| | 0.2% | **4.07** ±0.02 | **4.06** ±0.04 | **4.06** ±0.03 | **4.09** ±0.03 | **4.12** ±0.04 | **4.08** ±0.04 |
| | 0.5% | **4.08** ±0.03 | **4.11** ±0.03 | **4.08** ±0.03 | **4.12** ±0.03 | **4.08** ±0.03 | **4.11** ±0.03 |
| WRN-16-4 | 1% | **4.08** ±0.02 | **4.10** ±0.04 | **4.09** ±0.02 | **4.11** ±0.04 | **4.09** ±0.02 | **4.10** ±0.04 |
| (CIFAR10) | 2% | **4.13** ±0.03 | **4.12** ±0.02 | **4.14** ±0.03 | **4.13** ±0.02 | **4.13** ±0.04 | **4.12** ±0.02 |
| | 5% | 4.26 ±0.02 | **4.17** ±0.04 | 4.26 ±0.02 | **4.17** ±0.04 | 4.26 ±0.02 | **4.16** ±0.04 |
| | 10% | 4.35 ±0.03 | **4.28** ±0.03 | 4.34 ±0.03 | **4.27** ±0.02 | 4.35 ±0.03 | **4.27** ±0.03 |
| | 20% | 4.65 ±0.03 | **4.51** ±0.02 | 4.65 ±0.03 | **4.51** ±0.02 | 4.66 ±0.03 | **4.51** ±0.02 |
| | 1% | **19.18** ±0.18 | **19.10** ±0.24 | 31.27 ±8.11 | 25.36 ±6.19 | 25.22 ±6.11 | **19.14** ±0.23 |
| | 2% | **18.94** ±0.27 | **18.93** ±0.13 | **19.05** ±0.24 | **19.03** ±0.16 | **18.99** ±0.27 | **18.91** ±0.13 |
| GCN | 5% | **19.22** ±0.15 | **19.00** ±0.19 | **19.21** ±0.15 | **19.01** ±0.22 | **19.18** ±0.14 | **18.97** ±0.17 |
| (NCI1) | 10% | **19.18** ±0.18 | **19.03** ±0.18 | **19.26** ±0.17 | **19.25** ±0.14 | **19.23** ±0.17 | **19.03** ±0.19 |
| | 20% | **18.63** ±0.23 | **18.89** ±0.21 | **18.59** ±0.24 | **18.86** ±0.22 | **18.61** ±0.24 | **18.88** ±0.21 |
| | 50% | 20.77 ±0.32 | **18.86** ±0.16 | 20.83 ±0.34 | **18.89** ±0.22 | 20.82 ±0.32 | **18.83** ±0.16 |
| | 0.1% | **3.30** ±0.01 | **3.30** ±0.01 | **3.31** ±0.01 | **3.31** ±0.01 | **3.30** ±0.01 | **3.30** ±0.01 |
| | 0.2% | **3.31** ±0.01 | **3.30** ±0.01 | **3.31** ±0.01 | **3.31** ±0.01 | **3.31** ±0.01 | **3.30** ±0.01 |
| | 0.5% | **3.30** ±0.01 | **3.29** ±0.01 | **3.30** ±0.01 | **3.29** ±0.01 | **3.30** ±0.01 | **3.29** ±0.01 |
| MLP | 1% | **3.31** ±0.01 | **3.31** ±0.01 | **3.31** ±0.01 | **3.31** ±0.01 | **3.31** ±0.01 | **3.31** ±0.01 |
| (COVERTYPE) | 2% | 3.33 ±0.01 | **3.31** ±0.00 | 3.33 ±0.01 | **3.31** ±0.00 | 3.33 ±0.01 | **3.31** ±0.00 |
| | 5% | 3.35 ±0.01 | **3.33** ±0.01 | 3.35 ±0.01 | **3.33** ±0.01 | 3.36 ±0.01 | **3.33** ±0.01 |
| | 10% | 3.42 ±0.01 | **3.38** ±0.01 | 3.42 ±0.01 | **3.38** ±0.01 | 3.42 ±0.01 | **3.38** ±0.01 |
| | 20% | 3.55 ±0.02 | **3.41** ±0.01 | 3.55 ±0.02 | **3.41** ±0.01 | 3.55 ±0.02 | **3.41** ±0.01 |
| | 0.1% | **7.55** ±0.03 | **7.58** ±0.05 | **7.56** ±0.03 | **7.58** ±0.05 | **7.57** ±0.03 | **7.58** ±0.04 |
| | 0.2% | **7.60** ±0.03 | **7.60** ±0.04 | **7.61** ±0.04 | **7.59** ±0.04 | **7.60** ±0.03 | **7.58** ±0.05 |
| | 0.5% | **7.52** ±0.05 | **7.58** ±0.04 | **7.52** ±0.05 | **7.58** ±0.04 | **7.52** ±0.05 | **7.57** ±0.03 |
| BILSTM | 1% | **7.55** ±0.04 | **7.55** ±0.06 | **7.56** ±0.04 | **7.52** ±0.06 | **7.55** ±0.04 | **7.53** ±0.06 |
| (AG_NEWS) | 2% | **7.49** ±0.04 | **7.53** ±0.05 | **7.49** ±0.04 | **7.52** ±0.05 | **7.48** ±0.04 | **7.52** ±0.06 |
| | 5% | 7.53 ±0.06 | **7.47** ±0.05 | 7.53 ±0.06 | **7.46** ±0.03 | 7.54 ±0.06 | **7.47** ±0.04 |
| | 10% | 7.62 ±0.04 | **7.54** ±0.06 | 7.57 ±0.05 | **7.53** ±0.04 | 7.62 ±0.04 | **7.52** ±0.04 |
| | 20% | 7.89 ±0.07 | **7.66** ±0.06 | 7.88 ±0.08 | **7.62** ±0.09 | 7.89 ±0.06 | **7.64** ±0.09 |

Table 6: Ensemble temperature scaling results for NLL. Bold values are not statistically different from the best in each row (within 1.96 SEM).

| | | Ensemble: NLL ↓ | | | | | |
|---|---|---|---|---|---|---|---|
| | | No scaling | | Individual scaling | | Joint scaling | |
| Model / Dataset | Val. % | Shared | Overlapping | Shared | Overlapping | Shared | Overlapping |
| WRN-16-4 (CIFAR10) | 0.1% | **0.129** $_{\pm0.000}$ | **0.129** $_{\pm0.001}$ | 0.133 $_{\pm0.002}$ | 0.351 $_{\pm0.218}$ | 0.544 $_{\pm0.396}$ | 0.136 $_{\pm0.003}$ |
| | 0.2% | 0.130 $_{\pm0.001}$ | **0.127** $_{\pm0.001}$ | 0.131 $_{\pm0.002}$ | 0.130 $_{\pm0.001}$ | 1.052 $_{\pm0.911}$ | 0.129 $_{\pm0.001}$ |
| | 0.5% | **0.129** $_{\pm0.001}$ | **0.129** $_{\pm0.001}$ | 0.130 $_{\pm0.001}$ | 0.131 $_{\pm0.001}$ | **0.129** $_{\pm0.001}$ | **0.129** $_{\pm0.001}$ |
| | 1% | **0.129** $_{\pm0.000}$ | **0.129** $_{\pm0.000}$ | 0.131 $_{\pm0.001}$ | 0.131 $_{\pm0.001}$ | **0.129** $_{\pm0.001}$ | **0.129** $_{\pm0.001}$ |
| | 2% | 0.130 $_{\pm0.000}$ | **0.129** $_{\pm0.001}$ | 0.132 $_{\pm0.001}$ | 0.131 $_{\pm0.001}$ | **0.129** $_{\pm0.000}$ | **0.128** $_{\pm0.001}$ |
| | 5% | 0.134 $_{\pm0.000}$ | 0.132 $_{\pm0.001}$ | 0.136 $_{\pm0.001}$ | 0.134 $_{\pm0.001}$ | 0.133 $_{\pm0.000}$ | **0.131** $_{\pm0.001}$ |
| | 10% | 0.136 $_{\pm0.001}$ | 0.134 $_{\pm0.000}$ | 0.138 $_{\pm0.001}$ | 0.136 $_{\pm0.000}$ | 0.135 $_{\pm0.001}$ | **0.133** $_{\pm0.000}$ |
| | 20% | 0.145 $_{\pm0.001}$ | **0.142** $_{\pm0.001}$ | 0.147 $_{\pm0.001}$ | 0.145 $_{\pm0.001}$ | 0.144 $_{\pm0.001}$ | **0.141** $_{\pm0.001}$ |
| GCN (NCI1) | 1% | 0.443 $_{\pm0.003}$ | **0.440** $_{\pm0.001}$ | 0.505 $_{\pm0.034}$ | 0.473 $_{\pm0.025}$ | 0.480 $_{\pm0.025}$ | 0.452 $_{\pm0.004}$ |
| | 2% | **0.440** $_{\pm0.001}$ | **0.440** $_{\pm0.001}$ | 0.449 $_{\pm0.003}$ | 0.448 $_{\pm0.003}$ | 0.456 $_{\pm0.010}$ | 0.444 $_{\pm0.003}$ |
| | 5% | **0.440** $_{\pm0.002}$ | **0.439** $_{\pm0.003}$ | **0.443** $_{\pm0.002}$ | 0.451 $_{\pm0.004}$ | **0.443** $_{\pm0.003}$ | **0.443** $_{\pm0.002}$ |
| | 10% | **0.440** $_{\pm0.003}$ | **0.441** $_{\pm0.003}$ | 0.448 $_{\pm0.004}$ | 0.449 $_{\pm0.003}$ | **0.442** $_{\pm0.003}$ | **0.443** $_{\pm0.002}$ |
| | 20% | 0.449 $_{\pm0.004}$ | **0.434** $_{\pm0.003}$ | 0.449 $_{\pm0.002}$ | 0.444 $_{\pm0.002}$ | 0.443 $_{\pm0.002}$ | **0.434** $_{\pm0.002}$ |
| | 50% | 0.559 $_{\pm0.009}$ | 0.441 $_{\pm0.003}$ | 0.482 $_{\pm0.003}$ | 0.454 $_{\pm0.002}$ | 0.478 $_{\pm0.004}$ | **0.432** $_{\pm0.001}$ |
| MLP (COVERTYPE) | 0.1% | **0.085** $_{\pm0.000}$ | **0.085** $_{\pm0.000}$ | 0.085 $_{\pm0.000}$ | **0.084** $_{\pm0.000}$ | 0.085 $_{\pm0.000}$ | **0.084** $_{\pm0.000}$ |
| | 0.2% | 0.086 $_{\pm0.001}$ | 0.085 $_{\pm0.000}$ | 0.085 $_{\pm0.001}$ | **0.084** $_{\pm0.000}$ | 0.085 $_{\pm0.001}$ | **0.084** $_{\pm0.000}$ |
| | 0.5% | 0.085 $_{\pm0.000}$ | 0.085 $_{\pm0.000}$ | 0.085 $_{\pm0.000}$ | **0.084** $_{\pm0.000}$ | 0.085 $_{\pm0.000}$ | **0.084** $_{\pm0.000}$ |
| | 1% | 0.085 $_{\pm0.000}$ | 0.085 $_{\pm0.000}$ | **0.085** $_{\pm0.000}$ | 0.085 $_{\pm0.000}$ | **0.085** $_{\pm0.000}$ | **0.085** $_{\pm0.000}$ |
| | 2% | 0.085 $_{\pm0.000}$ | 0.085 $_{\pm0.000}$ | 0.085 $_{\pm0.000}$ | 0.085 $_{\pm0.000}$ | 0.085 $_{\pm0.000}$ | **0.084** $_{\pm0.000}$ |
| | 5% | 0.086 $_{\pm0.000}$ | 0.085 $_{\pm0.000}$ | 0.086 $_{\pm0.000}$ | **0.085** $_{\pm0.000}$ | 0.086 $_{\pm0.000}$ | **0.085** $_{\pm0.000}$ |
| | 10% | 0.088 $_{\pm0.000}$ | **0.087** $_{\pm0.000}$ | 0.088 $_{\pm0.000}$ | **0.086** $_{\pm0.000}$ | 0.087 $_{\pm0.000}$ | **0.086** $_{\pm0.000}$ |
| | 20% | 0.092 $_{\pm0.000}$ | 0.087 $_{\pm0.000}$ | 0.092 $_{\pm0.000}$ | 0.087 $_{\pm0.000}$ | 0.092 $_{\pm0.000}$ | **0.087** $_{\pm0.000}$ |
| BILSTM (AG_NEWS) | 0.1% | **0.235** $_{\pm0.001}$ | **0.235** $_{\pm0.001}$ | 0.250 $_{\pm0.004}$ | 0.246 $_{\pm0.002}$ | **0.235** $_{\pm0.001}$ | 0.238 $_{\pm0.001}$ |
| | 0.2% | **0.234** $_{\pm0.001}$ | **0.235** $_{\pm0.001}$ | 0.243 $_{\pm0.004}$ | 0.243 $_{\pm0.003}$ | **0.236** $_{\pm0.001}$ | 0.237 $_{\pm0.002}$ |
| | 0.5% | **0.234** $_{\pm0.001}$ | **0.233** $_{\pm0.001}$ | 0.241 $_{\pm0.002}$ | 0.241 $_{\pm0.003}$ | **0.233** $_{\pm0.001}$ | **0.233** $_{\pm0.001}$ |
| | 1% | 0.235 $_{\pm0.001}$ | 0.235 $_{\pm0.000}$ | 0.239 $_{\pm0.002}$ | 0.240 $_{\pm0.001}$ | 0.235 $_{\pm0.001}$ | **0.234** $_{\pm0.000}$ |
| | 2% | **0.234** $_{\pm0.001}$ | **0.234** $_{\pm0.000}$ | 0.239 $_{\pm0.002}$ | 0.242 $_{\pm0.001}$ | **0.234** $_{\pm0.001}$ | **0.235** $_{\pm0.000}$ |
| | 5% | **0.235** $_{\pm0.001}$ | **0.235** $_{\pm0.001}$ | 0.242 $_{\pm0.001}$ | 0.243 $_{\pm0.001}$ | **0.234** $_{\pm0.001}$ | **0.235** $_{\pm0.001}$ |
| | 10% | **0.236** $_{\pm0.001}$ | **0.236** $_{\pm0.001}$ | 0.244 $_{\pm0.001}$ | 0.246 $_{\pm0.001}$ | **0.236** $_{\pm0.001}$ | 0.238 $_{\pm0.001}$ |
| | 20% | 0.248 $_{\pm0.003}$ | **0.242** $_{\pm0.001}$ | 0.258 $_{\pm0.004}$ | 0.256 $_{\pm0.001}$ | 0.247 $_{\pm0.002}$ | 0.246 $_{\pm0.001}$ |

Table 7: Ensemble temperature scaling results for ECE. Bold values are not statistically different from the best in each row (within 1.96 SEM).

| | | Ensemble: ECE ↓ | | | | | |
| --- | --- | --- | --- | --- | --- | --- | --- |
| | | No scaling | | Individual scaling | | Joint scaling | |
| Model / Dataset | Val. % | Shared | Overlapping | Shared | Overlapping | Shared | Overlapping |
| WRN-16-4 (CIFAR10) | 0.1% | **0.009** ±0.000 | **0.009** ±0.000 | 0.012 ±0.003 | 0.019 ±0.009 | 0.012 ±0.002 | 0.010 ±0.002 |
| | 0.2% | **0.009** ±0.000 | **0.009** ±0.000 | **0.009** ±0.002 | 0.011 ±0.002 | **0.009** ±0.002 | **0.008** ±0.001 |
| | 0.5% | 0.009 ±0.000 | 0.009 ±0.000 | 0.011 ±0.001 | 0.011 ±0.001 | **0.006** ±0.000 | **0.007** ±0.001 |
| | 1% | 0.009 ±0.000 | 0.009 ±0.000 | 0.012 ±0.001 | 0.012 ±0.001 | **0.006** ±0.000 | 0.007 ±0.001 |
| | 2% | 0.009 ±0.000 | 0.009 ±0.000 | 0.012 ±0.001 | 0.012 ±0.001 | **0.006** ±0.000 | 0.007 ±0.001 |
| | 5% | 0.009 ±0.000 | 0.010 ±0.000 | 0.013 ±0.001 | 0.013 ±0.001 | **0.005** ±0.000 | 0.007 ±0.001 |
| | 10% | 0.008 ±0.000 | 0.010 ±0.000 | 0.012 ±0.000 | 0.013 ±0.000 | **0.005** ±0.000 | 0.007 ±0.000 |
| | 20% | 0.008 ±0.000 | 0.011 ±0.000 | 0.013 ±0.000 | 0.016 ±0.000 | **0.006** ±0.000 | 0.009 ±0.000 |
| GCN (NCI1) | 1% | 0.034 ±0.002 | **0.027** ±0.001 | 0.113 ±0.034 | 0.083 ±0.027 | 0.084 ±0.026 | 0.055 ±0.008 |
| | 2% | **0.033** ±0.002 | **0.033** ±0.002 | 0.053 ±0.004 | 0.063 ±0.006 | 0.049 ±0.007 | 0.052 ±0.006 |
| | 5% | **0.028** ±0.001 | **0.028** ±0.002 | 0.048 ±0.005 | 0.070 ±0.008 | 0.039 ±0.003 | 0.052 ±0.005 |
| | 10% | **0.034** ±0.003 | **0.034** ±0.001 | 0.065 ±0.007 | 0.068 ±0.005 | 0.047 ±0.006 | 0.054 ±0.004 |
| | 20% | 0.039 ±0.003 | **0.033** ±0.002 | 0.064 ±0.006 | 0.072 ±0.003 | 0.048 ±0.004 | 0.053 ±0.003 |
| | 50% | 0.086 ±0.003 | **0.043** ±0.002 | 0.057 ±0.003 | 0.092 ±0.002 | **0.041** ±0.003 | 0.052 ±0.003 |
| MLP (COVERTYPE) | 0.1% | 0.003 ±0.000 | 0.003 ±0.000 | 0.004 ±0.001 | **0.002** ±0.000 | 0.003 ±0.001 | **0.002** ±0.001 |
| | 0.2% | 0.004 ±0.001 | 0.003 ±0.000 | 0.003 ±0.001 | 0.002 ±0.000 | 0.003 ±0.000 | **0.001** ±0.000 |
| | 0.5% | 0.004 ±0.000 | 0.003 ±0.000 | **0.002** ±0.001 | **0.002** ±0.000 | 0.003 ±0.000 | **0.002** ±0.000 |
| | 1% | 0.003 ±0.000 | 0.003 ±0.000 | **0.002** ±0.000 | 0.002 ±0.000 | 0.002 ±0.000 | **0.002** ±0.000 |
| | 2% | 0.003 ±0.000 | 0.003 ±0.000 | 0.002 ±0.000 | 0.002 ±0.000 | 0.002 ±0.000 | **0.001** ±0.000 |
| | 5% | 0.003 ±0.000 | 0.003 ±0.000 | 0.002 ±0.000 | 0.002 ±0.000 | 0.002 ±0.000 | **0.001** ±0.000 |
| | 10% | 0.002 ±0.000 | 0.004 ±0.000 | 0.002 ±0.000 | 0.004 ±0.000 | **0.002** ±0.000 | 0.002 ±0.000 |
| | 20% | **0.002** ±0.000 | 0.004 ±0.000 | 0.002 ±0.000 | 0.005 ±0.000 | **0.001** ±0.000 | 0.003 ±0.000 |
| BILSTM (AG_NEWS) | 0.1% | **0.019** ±0.001 | **0.019** ±0.001 | 0.052 ±0.007 | 0.047 ±0.004 | 0.028 ±0.003 | 0.034 ±0.004 |
| | 0.2% | **0.018** ±0.001 | **0.019** ±0.001 | 0.041 ±0.008 | 0.040 ±0.007 | 0.021 ±0.004 | 0.029 ±0.004 |
| | 0.5% | **0.020** ±0.000 | **0.020** ±0.001 | 0.041 ±0.004 | 0.040 ±0.004 | 0.022 ±0.001 | 0.026 ±0.002 |
| | 1% | **0.020** ±0.001 | **0.020** ±0.001 | 0.036 ±0.002 | 0.039 ±0.001 | **0.020** ±0.001 | 0.026 ±0.001 |
| | 2% | **0.020** ±0.000 | 0.021 ±0.001 | 0.038 ±0.002 | 0.042 ±0.002 | 0.021 ±0.001 | 0.027 ±0.001 |
| | 5% | **0.021** ±0.001 | 0.023 ±0.001 | 0.042 ±0.002 | 0.043 ±0.001 | **0.021** ±0.001 | 0.028 ±0.001 |
| | 10% | **0.022** ±0.001 | **0.023** ±0.001 | 0.042 ±0.001 | 0.046 ±0.001 | **0.022** ±0.001 | 0.030 ±0.001 |
| | 20% | **0.028** ±0.004 | **0.027** ±0.001 | 0.051 ±0.005 | 0.056 ±0.001 | **0.026** ±0.001 | 0.037 ±0.001 |

Table 8: Average individual model temperature scaling results for Classification Error (%). Bold values are not statistically different from the best in each row (within 1.96 SEM).

| | | Average Individual: Classification Error (%) $\downarrow$ | | | | | |
|---|---|---|---|---|---|---|---|
| | | No scaling | | Individual scaling | | Joint scaling | |
| Model / Dataset | Val. % | Shared | Overlapping | Shared | Overlapping | Shared | Overlapping |
| WRN-16-4 (CIFAR10) | 0.1% | **4.77** $_{\pm 0.01}$ | 4.81 $_{\pm 0.02}$ | **4.77** $_{\pm 0.01}$ | 14.34 $_{\pm 9.52}$ | 4.78 $_{\pm 0.01}$ | 4.81 $_{\pm 0.02}$ |
| | 0.2% | 4.77 $_{\pm 0.02}$ | **4.72** $_{\pm 0.02}$ | 4.77 $_{\pm 0.02}$ | **4.72** $_{\pm 0.02}$ | 4.78 $_{\pm 0.02}$ | **4.72** $_{\pm 0.02}$ |
| | 0.5% | **4.79** $_{\pm 0.02}$ | **4.78** $_{\pm 0.03}$ | **4.79** $_{\pm 0.02}$ | **4.78** $_{\pm 0.03}$ | **4.79** $_{\pm 0.02}$ | **4.78** $_{\pm 0.03}$ |
| | 1% | **4.77** $_{\pm 0.01}$ | **4.76** $_{\pm 0.02}$ | 4.78 $_{\pm 0.01}$ | **4.76** $_{\pm 0.02}$ | 4.78 $_{\pm 0.01}$ | **4.76** $_{\pm 0.02}$ |
| | 2% | **4.80** $_{\pm 0.01}$ | **4.81** $_{\pm 0.02}$ | **4.80** $_{\pm 0.01}$ | **4.81** $_{\pm 0.02}$ | **4.80** $_{\pm 0.01}$ | **4.81** $_{\pm 0.02}$ |
| | 5% | **4.98** $_{\pm 0.02}$ | **4.93** $_{\pm 0.03}$ | **4.98** $_{\pm 0.02}$ | **4.93** $_{\pm 0.03}$ | **4.98** $_{\pm 0.02}$ | **4.92** $_{\pm 0.03}$ |
| | 10% | 5.04 $_{\pm 0.02}$ | **5.01** $_{\pm 0.02}$ | 5.04 $_{\pm 0.02}$ | **5.01** $_{\pm 0.02}$ | 5.04 $_{\pm 0.02}$ | **5.01** $_{\pm 0.02}$ |
| | 20% | **5.37** $_{\pm 0.02}$ | **5.34** $_{\pm 0.02}$ | **5.37** $_{\pm 0.02}$ | **5.34** $_{\pm 0.02}$ | **5.37** $_{\pm 0.02}$ | **5.34** $_{\pm 0.02}$ |
| GCN (NCI1) | 1% | **20.26** $_{\pm 0.15}$ | **20.04** $_{\pm 0.17}$ | 31.86 $_{\pm 7.85}$ | 27.21 $_{\pm 5.95}$ | 26.05 $_{\pm 5.88}$ | **20.05** $_{\pm 0.17}$ |
| | 2% | **20.04** $_{\pm 0.15}$ | **20.05** $_{\pm 0.15}$ | **20.06** $_{\pm 0.15}$ | **20.07** $_{\pm 0.15}$ | **20.05** $_{\pm 0.15}$ | **20.05** $_{\pm 0.15}$ |
| | 5% | **19.80** $_{\pm 0.14}$ | **19.87** $_{\pm 0.16}$ | **19.79** $_{\pm 0.14}$ | **19.87** $_{\pm 0.16}$ | **19.79** $_{\pm 0.14}$ | **19.86** $_{\pm 0.15}$ |
| | 10% | **19.96** $_{\pm 0.09}$ | **20.03** $_{\pm 0.14}$ | **19.98** $_{\pm 0.09}$ | **20.08** $_{\pm 0.13}$ | **19.97** $_{\pm 0.09}$ | **20.03** $_{\pm 0.14}$ |
| | 20% | **19.56** $_{\pm 0.13}$ | **19.70** $_{\pm 0.15}$ | **19.55** $_{\pm 0.13}$ | **19.70** $_{\pm 0.15}$ | **19.56** $_{\pm 0.13}$ | **19.70** $_{\pm 0.15}$ |
| | 50% | 21.66 $_{\pm 0.28}$ | **21.19** $_{\pm 0.14}$ | 21.67 $_{\pm 0.29}$ | **21.20** $_{\pm 0.14}$ | 21.67 $_{\pm 0.28}$ | **21.19** $_{\pm 0.13}$ |
| MLP (COVERTYPE) | 0.1% | **3.43** $_{\pm 0.01}$ | **3.42** $_{\pm 0.00}$ | **3.43** $_{\pm 0.01}$ | **3.42** $_{\pm 0.00}$ | **3.43** $_{\pm 0.01}$ | **3.42** $_{\pm 0.00}$ |
| | 0.2% | 3.51 $_{\pm 0.08}$ | **3.42** $_{\pm 0.00}$ | 3.51 $_{\pm 0.08}$ | **3.42** $_{\pm 0.00}$ | 3.51 $_{\pm 0.08}$ | **3.42** $_{\pm 0.00}$ |
| | 0.5% | 3.46 $_{\pm 0.03}$ | **3.42** $_{\pm 0.00}$ | 3.46 $_{\pm 0.03}$ | **3.42** $_{\pm 0.00}$ | 3.46 $_{\pm 0.03}$ | **3.42** $_{\pm 0.00}$ |
| | 1% | **3.44** $_{\pm 0.00}$ | 3.46 $_{\pm 0.03}$ | **3.44** $_{\pm 0.00}$ | 3.46 $_{\pm 0.03}$ | **3.44** $_{\pm 0.00}$ | 3.46 $_{\pm 0.03}$ |
| | 2% | 3.47 $_{\pm 0.04}$ | **3.45** $_{\pm 0.00}$ | 3.47 $_{\pm 0.04}$ | **3.45** $_{\pm 0.00}$ | 3.47 $_{\pm 0.04}$ | **3.45** $_{\pm 0.00}$ |
| | 5% | 3.49 $_{\pm 0.01}$ | **3.48** $_{\pm 0.01}$ | 3.49 $_{\pm 0.01}$ | **3.48** $_{\pm 0.01}$ | 3.49 $_{\pm 0.01}$ | **3.48** $_{\pm 0.01}$ |
| | 10% | **3.55** $_{\pm 0.01}$ | 3.59 $_{\pm 0.03}$ | **3.55** $_{\pm 0.01}$ | 3.59 $_{\pm 0.03}$ | **3.55** $_{\pm 0.01}$ | 3.59 $_{\pm 0.03}$ |
| | 20% | 3.68 $_{\pm 0.02}$ | **3.67** $_{\pm 0.01}$ | 3.69 $_{\pm 0.02}$ | **3.67** $_{\pm 0.01}$ | 3.69 $_{\pm 0.02}$ | **3.67** $_{\pm 0.01}$ |
| BILSTM (AG_NEWS) | 0.1% | **9.52** $_{\pm 0.05}$ | **9.54** $_{\pm 0.03}$ | **9.52** $_{\pm 0.05}$ | **9.54** $_{\pm 0.03}$ | **9.52** $_{\pm 0.05}$ | **9.54** $_{\pm 0.03}$ |
| | 0.2% | **9.53** $_{\pm 0.04}$ | **9.60** $_{\pm 0.06}$ | **9.53** $_{\pm 0.04}$ | **9.60** $_{\pm 0.07}$ | **9.53** $_{\pm 0.04}$ | **9.60** $_{\pm 0.06}$ |
| | 0.5% | **9.56** $_{\pm 0.03}$ | 9.66 $_{\pm 0.09}$ | **9.56** $_{\pm 0.03}$ | 9.66 $_{\pm 0.08}$ | **9.56** $_{\pm 0.03}$ | 9.66 $_{\pm 0.08}$ |
| | 1% | **9.57** $_{\pm 0.03}$ | **9.52** $_{\pm 0.03}$ | **9.58** $_{\pm 0.03}$ | **9.52** $_{\pm 0.03}$ | **9.57** $_{\pm 0.03}$ | **9.52** $_{\pm 0.03}$ |
| | 2% | **9.58** $_{\pm 0.04}$ | **9.62** $_{\pm 0.04}$ | **9.58** $_{\pm 0.04}$ | **9.62** $_{\pm 0.04}$ | **9.58** $_{\pm 0.04}$ | **9.62** $_{\pm 0.04}$ |
| | 5% | **9.67** $_{\pm 0.02}$ | **9.64** $_{\pm 0.06}$ | **9.67** $_{\pm 0.02}$ | **9.64** $_{\pm 0.06}$ | **9.67** $_{\pm 0.02}$ | **9.64** $_{\pm 0.06}$ |
| | 10% | **9.73** $_{\pm 0.04}$ | **9.73** $_{\pm 0.03}$ | **9.72** $_{\pm 0.04}$ | **9.73** $_{\pm 0.03}$ | **9.73** $_{\pm 0.04}$ | **9.73** $_{\pm 0.03}$ |
| | 20% | 10.48 $_{\pm 0.24}$ | **10.24** $_{\pm 0.03}$ | 10.47 $_{\pm 0.24}$ | **10.24** $_{\pm 0.03}$ | 10.48 $_{\pm 0.25}$ | **10.24** $_{\pm 0.03}$ |

Table 9: Average individual model temperature scaling results for NLL. Bold values are not statistically different from the best in each row (within 1.96 SEM).

| | | Average Individual: NLL ↓ | | | | | |
|---|---|---|---|---|---|---|---|
| | | No scaling | | Individual scaling | | Joint scaling | |
| Model / Dataset | Val. % | Shared | Overlapping | Shared | Overlapping | Shared | Overlapping |
| WRN-16-4 (CIFAR10) | 0.1% | **0.153** $_{\pm 0.001}$ | 0.155 $_{\pm 0.001}$ | 0.185 $_{\pm 0.023}$ | 0.438 $_{\pm 0.213}$ | 1.170 $_{\pm 0.966}$ | 0.178 $_{\pm 0.009}$ |
| | 0.2% | 0.155 $_{\pm 0.001}$ | **0.152** $_{\pm 0.001}$ | 0.162 $_{\pm 0.003}$ | 0.171 $_{\pm 0.013}$ | 2.528 $_{\pm 2.338}$ | 0.161 $_{\pm 0.004}$ |
| | 0.5% | **0.154** $_{\pm 0.001}$ | **0.154** $_{\pm 0.001}$ | **0.154** $_{\pm 0.001}$ | 0.156 $_{\pm 0.001}$ | 0.161 $_{\pm 0.002}$ | 0.158 $_{\pm 0.002}$ |
| | 1% | **0.154** $_{\pm 0.000}$ | **0.154** $_{\pm 0.000}$ | **0.154** $_{\pm 0.000}$ | **0.154** $_{\pm 0.000}$ | 0.161 $_{\pm 0.002}$ | 0.155 $_{\pm 0.001}$ |
| | 2% | **0.155** $_{\pm 0.001}$ | **0.154** $_{\pm 0.000}$ | **0.155** $_{\pm 0.000}$ | **0.154** $_{\pm 0.000}$ | 0.161 $_{\pm 0.002}$ | 0.156 $_{\pm 0.001}$ |
| | 5% | 0.161 $_{\pm 0.001}$ | **0.159** $_{\pm 0.001}$ | 0.161 $_{\pm 0.001}$ | **0.159** $_{\pm 0.001}$ | 0.167 $_{\pm 0.001}$ | 0.161 $_{\pm 0.001}$ |
| | 10% | 0.162 $_{\pm 0.001}$ | **0.161** $_{\pm 0.000}$ | 0.162 $_{\pm 0.001}$ | **0.161** $_{\pm 0.000}$ | 0.167 $_{\pm 0.001}$ | 0.162 $_{\pm 0.001}$ |
| | 20% | **0.173** $_{\pm 0.001}$ | **0.172** $_{\pm 0.001}$ | **0.172** $_{\pm 0.001}$ | **0.172** $_{\pm 0.001}$ | 0.178 $_{\pm 0.001}$ | 0.173 $_{\pm 0.001}$ |
| GCN (NCI1) | 1% | **0.474** $_{\pm 0.003}$ | **0.472** $_{\pm 0.003}$ | 0.523 $_{\pm 0.031}$ | 0.523 $_{\pm 0.021}$ | 0.513 $_{\pm 0.025}$ | 0.482 $_{\pm 0.007}$ |
| | 2% | **0.472** $_{\pm 0.002}$ | 0.477 $_{\pm 0.002}$ | 0.480 $_{\pm 0.010}$ | 0.474 $_{\pm 0.002}$ | 0.500 $_{\pm 0.022}$ | **0.469** $_{\pm 0.003}$ |
| | 5% | 0.472 $_{\pm 0.002}$ | 0.478 $_{\pm 0.004}$ | 0.465 $_{\pm 0.003}$ | 0.474 $_{\pm 0.005}$ | 0.473 $_{\pm 0.006}$ | **0.468** $_{\pm 0.003}$ |
| | 10% | 0.476 $_{\pm 0.004}$ | 0.493 $_{\pm 0.013}$ | 0.467 $_{\pm 0.003}$ | 0.479 $_{\pm 0.008}$ | **0.466** $_{\pm 0.002}$ | 0.476 $_{\pm 0.009}$ |
| | 20% | 0.493 $_{\pm 0.004}$ | 0.495 $_{\pm 0.004}$ | **0.468** $_{\pm 0.002}$ | **0.470** $_{\pm 0.003}$ | **0.470** $_{\pm 0.003}$ | **0.467** $_{\pm 0.003}$ |
| | 50% | 0.665 $_{\pm 0.012}$ | 0.634 $_{\pm 0.005}$ | 0.500 $_{\pm 0.004}$ | **0.491** $_{\pm 0.002}$ | 0.505 $_{\pm 0.004}$ | 0.496 $_{\pm 0.003}$ |
| MLP (COVERTYPE) | 0.1% | **0.088** $_{\pm 0.000}$ | **0.088** $_{\pm 0.000}$ | 0.088 $_{\pm 0.000}$ | 0.089 $_{\pm 0.000}$ | 0.089 $_{\pm 0.001}$ | 0.088 $_{\pm 0.000}$ |
| | 0.2% | 0.090 $_{\pm 0.002}$ | **0.088** $_{\pm 0.000}$ | 0.090 $_{\pm 0.002}$ | 0.088 $_{\pm 0.000}$ | 0.091 $_{\pm 0.002}$ | **0.088** $_{\pm 0.000}$ |
| | 0.5% | 0.089 $_{\pm 0.001}$ | **0.088** $_{\pm 0.000}$ | 0.089 $_{\pm 0.001}$ | **0.088** $_{\pm 0.000}$ | 0.090 $_{\pm 0.001}$ | 0.088 $_{\pm 0.000}$ |
| | 1% | **0.088** $_{\pm 0.000}$ | 0.089 $_{\pm 0.001}$ | **0.088** $_{\pm 0.000}$ | 0.089 $_{\pm 0.001}$ | 0.089 $_{\pm 0.000}$ | 0.089 $_{\pm 0.001}$ |
| | 2% | 0.089 $_{\pm 0.001}$ | **0.088** $_{\pm 0.000}$ | 0.089 $_{\pm 0.001}$ | **0.088** $_{\pm 0.000}$ | 0.090 $_{\pm 0.001}$ | **0.088** $_{\pm 0.000}$ |
| | 5% | 0.090 $_{\pm 0.000}$ | **0.089** $_{\pm 0.000}$ | 0.090 $_{\pm 0.000}$ | **0.089** $_{\pm 0.000}$ | 0.090 $_{\pm 0.000}$ | **0.089** $_{\pm 0.000}$ |
| | 10% | **0.091** $_{\pm 0.000}$ | 0.092 $_{\pm 0.001}$ | **0.091** $_{\pm 0.000}$ | 0.092 $_{\pm 0.001}$ | **0.091** $_{\pm 0.000}$ | 0.093 $_{\pm 0.001}$ |
| | 20% | 0.096 $_{\pm 0.000}$ | **0.095** $_{\pm 0.000}$ | 0.096 $_{\pm 0.000}$ | **0.095** $_{\pm 0.000}$ | 0.096 $_{\pm 0.000}$ | **0.095** $_{\pm 0.000}$ |
| BILSTM (AG_NEWS) | 0.1% | 0.323 $_{\pm 0.002}$ | 0.325 $_{\pm 0.001}$ | **0.289** $_{\pm 0.002}$ | 0.292 $_{\pm 0.001}$ | 0.295 $_{\pm 0.005}$ | **0.290** $_{\pm 0.002}$ |
| | 0.2% | 0.324 $_{\pm 0.002}$ | 0.323 $_{\pm 0.002}$ | **0.292** $_{\pm 0.002}$ | 0.295 $_{\pm 0.004}$ | 0.321 $_{\pm 0.015}$ | 0.302 $_{\pm 0.009}$ |
| | 0.5% | 0.324 $_{\pm 0.001}$ | 0.325 $_{\pm 0.003}$ | **0.288** $_{\pm 0.001}$ | 0.290 $_{\pm 0.002}$ | 0.309 $_{\pm 0.005}$ | 0.299 $_{\pm 0.003}$ |
| | 1% | 0.324 $_{\pm 0.001}$ | 0.322 $_{\pm 0.001}$ | **0.288** $_{\pm 0.001}$ | **0.286** $_{\pm 0.001}$ | 0.324 $_{\pm 0.008}$ | 0.292 $_{\pm 0.002}$ |
| | 2% | 0.321 $_{\pm 0.001}$ | 0.322 $_{\pm 0.001}$ | **0.288** $_{\pm 0.001}$ | 0.289 $_{\pm 0.001}$ | 0.319 $_{\pm 0.010}$ | 0.294 $_{\pm 0.001}$ |
| | 5% | 0.322 $_{\pm 0.001}$ | 0.321 $_{\pm 0.001}$ | **0.289** $_{\pm 0.001}$ | **0.289** $_{\pm 0.001}$ | 0.314 $_{\pm 0.003}$ | 0.296 $_{\pm 0.002}$ |
| | 10% | 0.321 $_{\pm 0.001}$ | 0.322 $_{\pm 0.001}$ | **0.292** $_{\pm 0.001}$ | **0.292** $_{\pm 0.001}$ | 0.319 $_{\pm 0.002}$ | 0.297 $_{\pm 0.001}$ |
| | 20% | 0.347 $_{\pm 0.006}$ | 0.339 $_{\pm 0.001}$ | 0.314 $_{\pm 0.008}$ | **0.307** $_{\pm 0.001}$ | 0.352 $_{\pm 0.015}$ | 0.313 $_{\pm 0.001}$ |

Table 10: Average individual model temperature scaling results for ECE. Bold values are not statistically different from the best in each row (within 1.96 SEM).

| Model / Dataset | Val. % | No scaling | | Individual scaling | | Joint scaling | |
|---|---|---|---|---|---|---|---|
| | | **Average Individual: ECE ↓** | | | | | |
| | | Shared | Overlapping | Shared | Overlapping | Shared | Overlapping |
| WRN-16-4 (CIFAR10) | 0.1% | **0.011** $_{\pm 0.000}$ | **0.011** $_{\pm 0.000}$ | 0.015 $_{\pm 0.002}$ | 0.026 $_{\pm 0.008}$ | 0.023 $_{\pm 0.004}$ | 0.020 $_{\pm 0.003}$ |
| | 0.2% | 0.011 $_{\pm 0.000}$ | **0.010** $_{\pm 0.000}$ | 0.014 $_{\pm 0.001}$ | 0.014 $_{\pm 0.001}$ | 0.023 $_{\pm 0.003}$ | 0.015 $_{\pm 0.002}$ |
| | 0.5% | **0.011** $_{\pm 0.000}$ | **0.011** $_{\pm 0.000}$ | **0.010** $_{\pm 0.001}$ | **0.011** $_{\pm 0.001}$ | 0.016 $_{\pm 0.001}$ | 0.014 $_{\pm 0.001}$ |
| | 1% | 0.011 $_{\pm 0.000}$ | 0.010 $_{\pm 0.000}$ | 0.010 $_{\pm 0.001}$ | **0.009** $_{\pm 0.000}$ | 0.017 $_{\pm 0.001}$ | 0.012 $_{\pm 0.001}$ |
| | 2% | 0.011 $_{\pm 0.000}$ | 0.011 $_{\pm 0.000}$ | **0.009** $_{\pm 0.000}$ | **0.009** $_{\pm 0.001}$ | 0.016 $_{\pm 0.001}$ | 0.013 $_{\pm 0.001}$ |
| | 5% | 0.012 $_{\pm 0.000}$ | 0.012 $_{\pm 0.000}$ | **0.009** $_{\pm 0.000}$ | **0.010** $_{\pm 0.000}$ | 0.017 $_{\pm 0.000}$ | 0.014 $_{\pm 0.000}$ |
| | 10% | 0.011 $_{\pm 0.000}$ | 0.011 $_{\pm 0.000}$ | **0.009** $_{\pm 0.000}$ | **0.009** $_{\pm 0.000}$ | 0.016 $_{\pm 0.000}$ | 0.013 $_{\pm 0.000}$ |
| | 20% | 0.012 $_{\pm 0.000}$ | 0.013 $_{\pm 0.000}$ | **0.010** $_{\pm 0.000}$ | **0.010** $_{\pm 0.000}$ | 0.017 $_{\pm 0.000}$ | 0.014 $_{\pm 0.000}$ |
| GCN (NCI1) | 1% | **0.040** $_{\pm 0.001}$ | **0.039** $_{\pm 0.001}$ | 0.111 $_{\pm 0.032}$ | 0.101 $_{\pm 0.023}$ | 0.085 $_{\pm 0.025}$ | 0.061 $_{\pm 0.006}$ |
| | 2% | **0.039** $_{\pm 0.001}$ | 0.041 $_{\pm 0.001}$ | 0.054 $_{\pm 0.005}$ | 0.063 $_{\pm 0.004}$ | 0.054 $_{\pm 0.008}$ | 0.050 $_{\pm 0.004}$ |
| | 5% | **0.039** $_{\pm 0.001}$ | 0.041 $_{\pm 0.001}$ | 0.050 $_{\pm 0.003}$ | 0.065 $_{\pm 0.007}$ | 0.046 $_{\pm 0.003}$ | 0.049 $_{\pm 0.003}$ |
| | 10% | **0.043** $_{\pm 0.002}$ | **0.044** $_{\pm 0.001}$ | 0.058 $_{\pm 0.006}$ | 0.063 $_{\pm 0.004}$ | **0.046** $_{\pm 0.004}$ | 0.048 $_{\pm 0.002}$ |
| | 20% | 0.052 $_{\pm 0.002}$ | 0.052 $_{\pm 0.001}$ | 0.056 $_{\pm 0.005}$ | 0.061 $_{\pm 0.002}$ | **0.045** $_{\pm 0.002}$ | **0.045** $_{\pm 0.002}$ |
| | 50% | 0.112 $_{\pm 0.004}$ | 0.102 $_{\pm 0.001}$ | 0.054 $_{\pm 0.003}$ | 0.060 $_{\pm 0.002}$ | **0.045** $_{\pm 0.002}$ | **0.046** $_{\pm 0.001}$ |
| MLP (COVERTYPE) | 0.1% | **0.001** $_{\pm 0.000}$ | **0.001** $_{\pm 0.000}$ | 0.004 $_{\pm 0.001}$ | 0.004 $_{\pm 0.000}$ | 0.004 $_{\pm 0.001}$ | 0.004 $_{\pm 0.001}$ |
| | 0.2% | 0.002 $_{\pm 0.000}$ | **0.001** $_{\pm 0.000}$ | 0.003 $_{\pm 0.000}$ | 0.003 $_{\pm 0.000}$ | 0.006 $_{\pm 0.000}$ | 0.002 $_{\pm 0.000}$ |
| | 0.5% | 0.001 $_{\pm 0.000}$ | **0.001** $_{\pm 0.000}$ | 0.003 $_{\pm 0.000}$ | 0.002 $_{\pm 0.000}$ | 0.005 $_{\pm 0.001}$ | 0.003 $_{\pm 0.001}$ |
| | 1% | **0.001** $_{\pm 0.000}$ | **0.001** $_{\pm 0.000}$ | 0.002 $_{\pm 0.000}$ | 0.002 $_{\pm 0.000}$ | 0.005 $_{\pm 0.001}$ | 0.003 $_{\pm 0.000}$ |
| | 2% | 0.001 $_{\pm 0.000}$ | **0.001** $_{\pm 0.000}$ | 0.002 $_{\pm 0.000}$ | 0.002 $_{\pm 0.000}$ | 0.005 $_{\pm 0.001}$ | 0.003 $_{\pm 0.000}$ |
| | 5% | **0.001** $_{\pm 0.000}$ | **0.001** $_{\pm 0.000}$ | 0.002 $_{\pm 0.000}$ | 0.002 $_{\pm 0.000}$ | 0.004 $_{\pm 0.001}$ | 0.003 $_{\pm 0.000}$ |
| | 10% | **0.001** $_{\pm 0.000}$ | 0.002 $_{\pm 0.000}$ | **0.002** $_{\pm 0.000}$ | 0.002 $_{\pm 0.000}$ | 0.004 $_{\pm 0.000}$ | 0.003 $_{\pm 0.000}$ |
| | 20% | **0.002** $_{\pm 0.000}$ | 0.002 $_{\pm 0.000}$ | **0.002** $_{\pm 0.000}$ | **0.002** $_{\pm 0.000}$ | 0.004 $_{\pm 0.000}$ | 0.004 $_{\pm 0.000}$ |
| BILSTM (AG_NEWS) | 0.1% | 0.040 $_{\pm 0.000}$ | 0.040 $_{\pm 0.000}$ | 0.025 $_{\pm 0.003}$ | 0.024 $_{\pm 0.002}$ | 0.024 $_{\pm 0.004}$ | **0.019** $_{\pm 0.002}$ |
| | 0.2% | 0.040 $_{\pm 0.000}$ | 0.040 $_{\pm 0.001}$ | **0.024** $_{\pm 0.003}$ | **0.024** $_{\pm 0.002}$ | 0.036 $_{\pm 0.006}$ | **0.026** $_{\pm 0.005}$ |
| | 0.5% | 0.040 $_{\pm 0.000}$ | 0.040 $_{\pm 0.001}$ | **0.018** $_{\pm 0.001}$ | **0.018** $_{\pm 0.001}$ | 0.033 $_{\pm 0.003}$ | 0.026 $_{\pm 0.002}$ |
| | 1% | 0.040 $_{\pm 0.000}$ | 0.039 $_{\pm 0.000}$ | 0.018 $_{\pm 0.001}$ | **0.017** $_{\pm 0.000}$ | 0.039 $_{\pm 0.003}$ | 0.023 $_{\pm 0.001}$ |
| | 2% | 0.039 $_{\pm 0.001}$ | 0.039 $_{\pm 0.001}$ | 0.018 $_{\pm 0.001}$ | **0.016** $_{\pm 0.000}$ | 0.037 $_{\pm 0.003}$ | 0.023 $_{\pm 0.001}$ |
| | 5% | 0.038 $_{\pm 0.000}$ | 0.038 $_{\pm 0.001}$ | **0.016** $_{\pm 0.000}$ | **0.016** $_{\pm 0.000}$ | 0.035 $_{\pm 0.001}$ | 0.023 $_{\pm 0.001}$ |
| | 10% | 0.037 $_{\pm 0.001}$ | 0.038 $_{\pm 0.001}$ | **0.017** $_{\pm 0.000}$ | **0.017** $_{\pm 0.000}$ | 0.036 $_{\pm 0.001}$ | 0.022 $_{\pm 0.000}$ |
| | 20% | 0.040 $_{\pm 0.000}$ | 0.039 $_{\pm 0.000}$ | **0.019** $_{\pm 0.001}$ | **0.019** $_{\pm 0.001}$ | 0.041 $_{\pm 0.002}$ | 0.024 $_{\pm 0.000}$ |

## F    Supplementary Results for Early Stopping

This section shows the early stopping performance for the average of the individual models within the ensemble. As shown in Figure 8, the extended training duration under joint stopping can lead to a degradation in the performance and calibration of the individual models, even as the final ensemble's performance (shown in Figure 3) improves. This highlights the *ensemble optimality gap*, where the best strategy for the ensemble is not necessarily the best for its individual components.

## G    Interaction Between Early Stopping and Temperature Scaling

To assess whether temperature scaling offers an additive benefit after joint early stopping, we performed a follow-up experiment using a 5% validation split across 5 random seeds. A single shared holdout set was used first to determine the stopping point for the ensemble via joint early stopping, and then the same set was used to optimize the temperature for post-hoc calibration. We compare the performance of full ensembles ($M = 4$ for WRN and GCN; $M = 8$ for MLP and BiLSTM) before and after applying temperature scaling. The results in Table 11 show no significant or consistent improvement, suggesting that for these well-regularized models, further post-hoc calibration has a negligible effect.

Table 11: Test performance of ensembles regularized with Joint Early Stopping on a shared holdout, comparing results with and without subsequent Joint Temperature Scaling. The experiment was run with a 5% validation set. The results show no significant additive benefit. Bold values indicate the best performance for each metric and model (within 1.96 SEM).

| Model | Metric | No Scaling | Joint Scaling |
|---|---|---|---|
| GCN (NCI1) | Classification Error (%) ↓ | **20.71** $_{\pm 0.38}$ | **20.68** $_{\pm 0.39}$ |
| | NLL ↓ | **0.4729** $_{\pm 0.0046}$ | **0.4755** $_{\pm 0.0027}$ |
| | ECE ↓ | **0.0402** $_{\pm 0.0047}$ | **0.0395** $_{\pm 0.0029}$ |
| BILSTM (AG_NEWS) | Classification Error (%) ↓ | **7.60** $_{\pm 0.08}$ | **7.59** $_{\pm 0.07}$ |
| | NLL ↓ | **0.2406** $_{\pm 0.0009}$ | **0.2392** $_{\pm 0.0010}$ |
| | ECE ↓ | **0.0257** $_{\pm 0.0004}$ | 0.0273 $_{\pm 0.0006}$ |
| MLP (COVERTYPE) | Classification Error (%) ↓ | **3.23** $_{\pm 0.01}$ | **3.23** $_{\pm 0.01}$ |
| | NLL ↓ | **0.0849** $_{\pm 0.0001}$ | **0.0849** $_{\pm 0.0002}$ |
| | ECE ↓ | 0.0046 $_{\pm 0.0005}$ | **0.0031** $_{\pm 0.0002}$ |
| WRN-16-4 (CIFAR10) | Classification Error (%) ↓ | **6.83** $_{\pm 0.18}$ | **6.79** $_{\pm 0.19}$ |
| | NLL ↓ | 0.2233 $_{\pm 0.0026}$ | **0.2133** $_{\pm 0.0048}$ |
| | ECE ↓ | **0.0149** $_{\pm 0.0008}$ | 0.0188 $_{\pm 0.0017}$ |

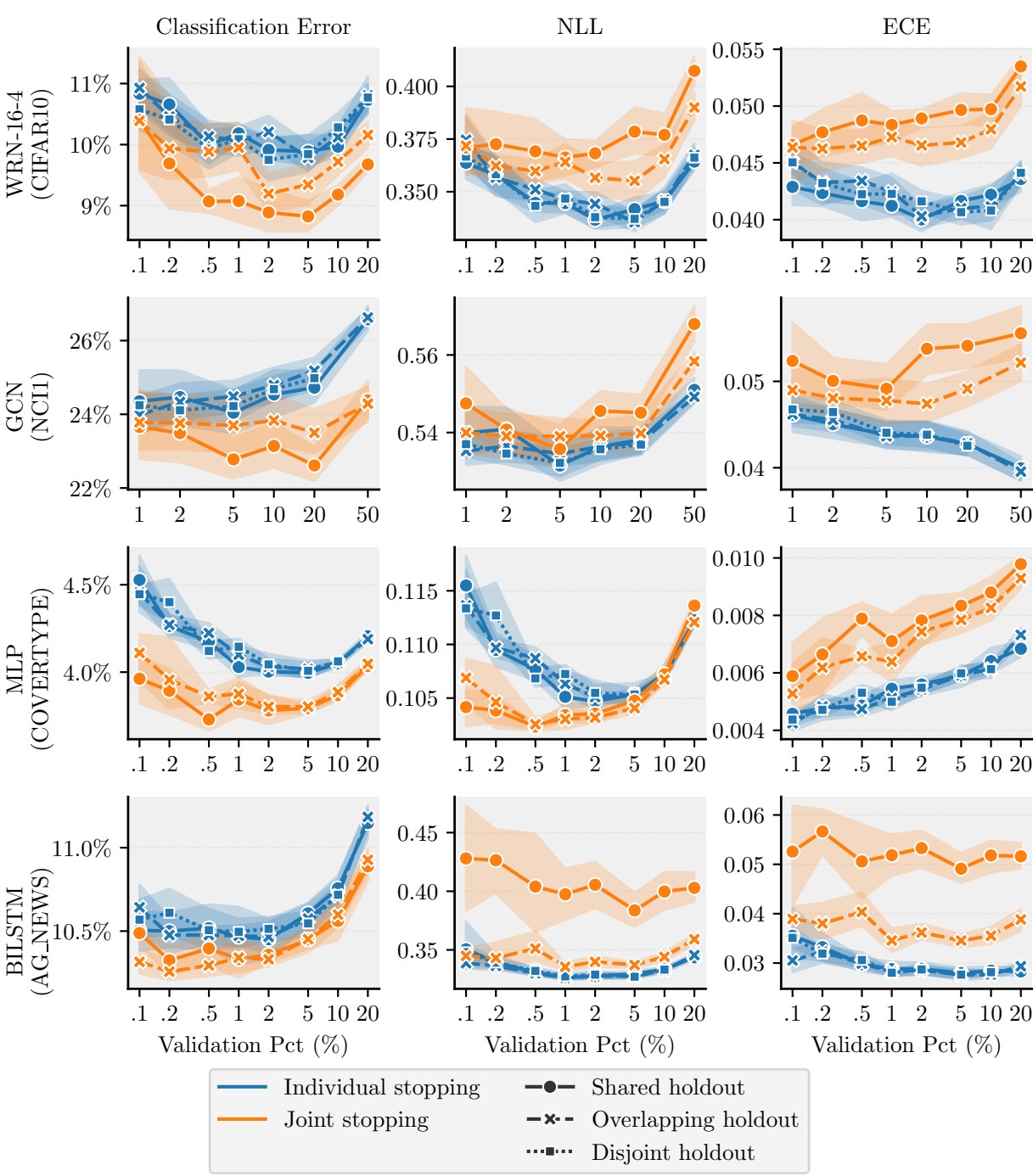

Figure 8: Test performance for the average of individual models within the ensemble, comparing different early stopping strategies. This figure shows results corresponding to the ensemble performance presented in Figure 3. (WRN: Wide ResNet; GCN: Graph Convolutional Network; MLP: Multi-Layer Perceptron; BILSTM: Bidirectional Long Short-Term Memory; NLL: negative log-likelihood; ECE: expected calibration error).

