# OpenReview forum: "On Joint Regularization and Calibration in Deep Ensembles"
_TMLR — Accepted by TMLR_

### Review · Reviewer_3Apb · 2025-05-16

**Summary Of Contributions:**

The paper studies how to improve the performance and calibration of deep ensembles by tuning key procedures such as weight decay, temperature scaling, and early stopping at the ensemble level, rather than treating each model separately before finally ensembling them. It shows that tuning based on ensemble validation performance leads to more accurate and better-calibrated predictions. The authors also propose an overlapping validation strategy that allows for joint evaluation while making efficient use of available data. Their experiments demonstrate that ensemble-aware tuning is effective and practical (particularly in low-data settings). The paper also shows that initialization choices in parameter-efficient methods like BatchEnsemble can significantly affect performance and calibration.

**Audience:**

Yes

**Broader Impact Concerns:**

No concern.

**Claims And Evidence:**

Yes

**Requested Changes:**

* The authors demonstrate that both joint temperature scaling and joint early stopping lead to improved ensemble performance. However, it is not entirely clear whether they recommend applying both techniques simultaneously. For example, training longer through joint early stopping may lead individual models to become slightly overconfident, which is acceptable due to the averaging effect of the ensemble. But if ensemble-level temperature scaling is applied afterward, is the benefit from longer training still necessary? It would be helpful if the authors clarified whether using both methods together provides additional value, and if so, how they recommend combining them in practice.

* The ensemble size is fixed at K=4, which is relatively small. Since larger ensembles are commonly used in practice to improve stability and robustness, it would be helpful if the authors could comment on whether they expect their findings to generalize to larger K. If possible, running at least a limited experiment with a larger ensemble size, or providing justification for the chosen value, would strengthen the paper.

* Investigations of temperature scaling at the ensemble learning is investigated in [1], with several variants and metrics. It would be interesting for the authors to discuss and possibly mention how their findings complements the recommendations described in [1].

* The authors are encouraged to expand the experimental evaluation to include additional model–dataset pairs. Given the empirical focus of the work, broader coverage would help demonstrate the robustness and generality of the reported findings. Even a small number of additional settings could strengthen the paper’s contributions and improve its practical relevance.

[1] Rahaman R, Thiery AH. Uncertainty quantification and deep ensembles. Advances in neural information processing systems. 2021

**Strengths And Weaknesses:**

Strengths:
-----------

* The paper addresses a practical and important problem in ensemble learning.
* It is commendable that the authors focus on empirical findings rather than introducing unnecessary theoretical results. The decision to let practical experiments drive the contributions is well-justified.
* Experiments are well expecuted
* The proposed overlapping holdout strategy is simple, effective, and motivated by real-world constraints.
* The study highlights a relatively under-explored issues such as the ensemble optimality gap and initialization sensitivity in BatchEnsemble.

Weaknesses:
-----------
* The study is based on only two model–dataset pairs, which limits the generality of the findings. Given the purely empirical nature of the work, a broader set of architectures and datasets would strengthen the conclusions.
* The ensemble size is fixed at K=4, which is relatively small. In practice, ensembles often use larger K to achieve more stable performance, so it would be valuable to explore whether the reported trends hold for larger ensembles.

---

> ### Author Response · Authors · 2025-08-08
>
> - The study is based on only two model–dataset pairs, which limits the generality of the findings. Given the purely empirical nature of the work, a broader set of architectures and datasets would strengthen the conclusions.
>
> As recommended, we have expanded our experimental evaluation and the revised paper now includes four diverse model/dataset benchmarks.
>
> - The ensemble size is fixed at K=4, which is relatively small. In practice, ensembles often use larger K to achieve more stable performance, so it would be valuable to explore whether the reported trends hold for larger ensembles.
>
> Our new experiments on the Covertype and AG News datasets use ensembles up to size M=8. While there are diminishing returns to adding more ensemble members, especially given the high computational cost, we believe this addition strengthens the relevance of our findings.
>
> - The authors demonstrate that both joint temperature scaling and joint early stopping lead to improved ensemble performance. However, it is not entirely clear whether they recommend applying both techniques simultaneously. For example, training longer through joint early stopping may lead individual models to become slightly overconfident, which is acceptable due to the averaging effect of the ensemble. But if ensemble-level temperature scaling is applied afterward, is the benefit from longer training still necessary? It would be helpful if the authors clarified whether using both methods together provides additional value, and if so, how they recommend combining them in practice.
>
> Great question. We have performed a new experiment to investigate this, and the results are included in Appendix G. In our settings, we did not find a significant additive benefit to applying temperature scaling after joint early stopping.
>
> - The ensemble size is fixed at K=4, which is relatively small. Since larger ensembles are commonly used in practice to improve stability and robustness, it would be helpful if the authors could comment on whether they expect their findings to generalize to larger K. If possible, running at least a limited experiment with a larger ensemble size, or providing justification for the chosen value, would strengthen the paper.
>
> Addressed above.
>
> - Investigations of temperature scaling at the ensemble learning is investigated in [1], with several variants and metrics. It would be interesting for the authors to discuss and possibly mention how their findings complements the recommendations described in [1].
>
> Thank you for this valuable suggestion. We have now cited this work and have included a direct comparison in Appendix D. Our approach is analogous to method (C) in their paper, and like them, we find that their methods (C) and (D) yield very similar results.
>
> - The authors are encouraged to expand the experimental evaluation to include additional model–dataset pairs. Given the empirical focus of the work, broader coverage would help demonstrate the robustness and generality of the reported findings. Even a small number of additional settings could strengthen the paper’s contributions and improve its practical relevance.
>
> As detailed above, we have included two additional models/datasets.
>
> [1] Rahaman R, Thiery AH. Uncertainty quantification and deep ensembles. Advances in neural information processing systems. 2021

---

### Review · Reviewer_BWK3 · 2025-06-04

**Summary Of Contributions:**

This paper proposed an approach to tune the ensemble models for regularization and calibration: Optimizing weight decay, temperature scaling, early stopping on the ensemble jointly instead of individually on each model. The authors tried to evaluated the ensemble optimality gap by comparing the performance with the baseline approach. Experiments show that on CIFAR 10 and NCI, the proposed approach leads to improved NLL, ECE and classification error. In addition, the authors also proposed "partially overlapping holdout" where each model uses two parts of validation data that is shared with its two "neighboring" models respectively. This aimed at reaching a balance between shared holdout and overlapping holdout.

**Audience:**

Yes

**Claims And Evidence:**

Yes

**Requested Changes:**

1.Revise Method section:
* Reframe the Methods section to explicitly define the joint optimization strategy, possibly as a formal algorithm or framework. Highlight how it improves upon conventional independent tuning. Add rigorous analysis. Carefully construct algorithms or rigor methodologies.

* Provide more rigorous justification of "partially overlapping holdout", and analyze its impact of invalidating full ensemble evaluation.
2.  Strengthen experiments: conduct experiments on more and larger datasets (e.g. ImageNet). Add more quantitative analysis.
3. The author can also discuss the limitation of jointly optimizing the ensembles. Tuning on individual models has practical benefits.

See "Strengths And Weaknesses" for more details.

**Strengths And Weaknesses:**

Strengths:



The paper offers a thorough and well-organized review of prior work in ensemble tuning, calibration, and validation strategies.

The empirical results suggest that jointly optimizing ensemble-level hyperparameters can lead to improved calibration and generalization, especially in resource-constrained settings.

Weakness:
1. Limited technical novelty
* The primary techniques used—weight decay, temperature scaling, early stopping, etc—are well-established. The idea of applying them jointly at the ensemble level is sensible but constitutes an incremental rather than conceptual innovation.

* The "partially overlapping holdout" has limited gains on data utilization but a notable drawback: it does not allow estimation on the entire ensemble but only pair-wise estimations. There is a simple alternative is to use m+1 fold validation dataset (by holding out 1 fold as D_eval for the full ensemble and using the other m folds for m individual models). This way, we can do full ensemble validation.

2. Insufficient Experimental Coverage:.
* The author only performed experiments on two datasets, CIFAR10 and NCI. Both are very small datasets. This cannot support the claim that "jointly tuning the ensemble matches or improves performance across all conditions". Experiments on more and larger datasets are expected.
* It relies on visual inspection and without statistical testing or quantitative analysis.

3. Method section lacks rigor.
* The current method section reads more like an experiment setup section. It lacks algorithmic or theoretical formalism that would clarify the generalizability of the proposed framework.

* Further methodological justification is needed for the design and effectiveness of the partially overlapping holdout.

---

> ### Author Response · Authors · 2025-08-08
>
> - The primary techniques used—weight decay, temperature scaling, early stopping, etc—are well-established. The idea of applying them jointly at the ensemble level is sensible but constitutes an incremental rather than conceptual innovation.
>
> We agree that our work is primarily an empirical investigation rather than a novel conceptual innovation. Our main contribution is to shine light on an underexplored aspect of deep ensemble training, as empirical studies in this specific area have been missing.
>
> - The "partially overlapping holdout" has limited gains on data utilization but a notable drawback: it does not allow estimation on the entire ensemble but only pair-wise estimations. There is a simple alternative is to use m+1 fold validation dataset (by holding out 1 fold as D_eval for the full ensemble and using the other m folds for m individual models). This way, we can do full ensemble validation.
>
> This is an interesting point; however, the suggested m+1 fold approach has the same primary drawback as a shared holdout: A portion of the available data is never used to train any of the models. Our overlapping holdout strategy is designed to make a specific trade-off: it ensures that all non-test data is used for training at least for some members of the ensemble, while still permitting joint evaluation of pairs, which we argue is a better proxy for full ensemble performance than individual model performance. We have clarified this trade-off in Section 3.3.
>
> - The author only performed experiments on two datasets, CIFAR10 and NCI. Both are very small datasets. This cannot support the claim that "jointly tuning the ensemble matches or improves performance across all conditions". Experiments on more and larger datasets are expected.
>
> We have significantly increased the number of experiments and now include results from four diverse benchmarks.
>
> - It relies on visual inspection and without statistical testing or quantitative analysis.
>
> To address this, we have substituted or added tables for key results including error bars where meaningful, as well as improved the visual presentation of figures to aid comparison.
>
> - The current method section reads more like an experiment setup section. It lacks algorithmic or theoretical formalism that would clarify the generalizability of the proposed framework.
>
> We have added a new subsection under Methodology where we formalize the individual/joint approach to model selection, to ensure that the framework is clearly defined.
>
> - Further methodological justification is needed for the design and effectiveness of the partially overlapping holdout.
>
> We have revised Section 3.3 to better highlight the trade-offs we are addressing with this strategy and to provide a stronger justification for its design.
>
> - Reframe the Methods section to explicitly define the joint optimization strategy, possibly as a formal algorithm or framework. Highlight how it improves upon conventional independent tuning. Add rigorous analysis. Carefully construct algorithms or rigor methodologies.
>
> We have added the new formal framework in Section 3.1. We chose to formulate the central criteria mathematically rather than presenting them as algorithms, since the procedures are not novel. We felt this approach better preserves the simplicity of the comparisons and aligns with the empirical focus of our study.
>
> - Provide more rigorous justification of "partially overlapping holdout", and analyze its impact of invalidating full ensemble evaluation.
>
> Adressed in the revised Section 3.3 as mentioned above.
>
> - Strengthen experiments: conduct experiments on more and larger datasets (e.g. ImageNet). Add more quantitative analysis.
>
> While the extreme computational cost of our study made ImageNet infeasible, we have added other large-scale datasets (Covertype and AG News) to strengthen our experiments.
>
> - The author can also discuss the limitation of jointly optimizing the ensembles. Tuning on individual models has practical benefits.
>
> We have added a new paragraph to the discussion in Section 5 that explicitly discusses this trade-off, acknowledging the practical benefits of the simpler, individual approach.

---

### Review · Reviewer_YiaQ · 2025-07-07

**Summary Of Contributions:**

The authors discuss hyperparameter optimisation in deep ensembles, with a focus on the resulting negative log likelihood (NLL), expected calibration error (ECE) and accuracy. For this, the authors provide ablations for weight decay, temperature scaling, and early stopping in deep ensembles. Additionally, they examine the initialisation of *fast weights* in BatchEnsembles, which aims to amortise the members' weights by reconstructing them from a shared weight matrix and local fast weights. The results for deep ensembles are generally in favour of joint optimisation strategies, although improvements are very mild, and show that BatchEnsembles are brittle concerning the initialisation strategy. Finally, the authors provide a set of *best practices* for hyperparameter tuning in ensembles.

**Audience:**

Yes

**Broader Impact Concerns:**

There are no ethical implications of the work.

**Claims And Evidence:**

No

**Requested Changes:**

- Section 4.4 needs better integration into the paper, and I would generally suggest overhauling the paper's structure to achieve better integration of the ablation.
- The results are currently rather limited, and additional ablations (using more models and datasets) would help strengthen the points.
- As the current hyperparameter ablations show rather mild improvements at best, I believe that it would be necessary to tone down the claims in the submission and explore why those improvements are so mild.

**Strengths And Weaknesses:**

# Strengths
The paper is well-written and easy to follow. Moreover, it provides an evaluation that, at least for BatchEnsembles, is somewhat insightful. The results are well-presented, and the ablations appear to be well-executed.

# Weaknesses
I believe the main weakness of this work is that the insights gained aren't particularly convincing, except for the experiments on BatchEnsembles. This is unfortunate as the work is otherwise well executed, and while there are some mild differences to be observed, it raises the question if they are sufficient to give strong arguments for either the joint objectives or the individual ones. I will discuss this in more detail below. Moreover, as also pointed out by the authors, the current study is very limited in scope regarding the model classes (only two models) and datasets (only two datasets). This potentially further limits the conclusions that can be drawn from the experiments.

# Detailed Comments
- General, it would be helpful to add equation numbers to all equations.
- The authors state that the paper examines the impact of hyperparameters on the *ensemble optimality gap*. However, it is never clearly stated what this optimality gap is or how it may be formalised.
- Page 5, typo: "objective: should" -> "objective: Should"
- I think it would help to formalise the two approaches (e.g., page 7) that are evaluated against, meaning the performance of an ensemble (equation on page 2) and the average performance of individual members. The latter is clearly a less meaningful metric as it does not correspond to any sensible probabilistic model.
- I did like the results discussed in 4.1, but found that the benefits obtained through shared weight-decay to be marginal, even though the authors state *strong benefits*, which makes me think that this is overselling of the actual improvements (difference between 0.008 and 0.006 in ECE).
- Similarly, I found the improvements in 4.2 to be modest at best, as accuracy and NLL are essentially equivalent for the ensemble, and only slight improvements in ECE can be observed.
- Again, same story for 4.3. Here, I think the authors again oversell the improvements and even the statement that one can find "joint stopping generally outperforms disjoint holdouts" is overselling at best. At least to my eye, there is no visible improvement or difference at all.
- I was slightly confused by the wording on page 12, saying "overall ensemble's validation performance". Is the performance showing the validation performance or the test performance in the figures of the paper? I believe it should be the latter.
- The ablation on BatchEnsembles is somewhat ad hoc and doesn't really connect well to the rest. Unfortunately for the rest, those results seem to be the most interesting ones. I believe this section would need better integration into the rest of the paper.
- The ablation on BatchEnsembles is somewhat too short, considering it reveals some patterns while the other ablations aren't very insightful. It would be interesting to study the effect of initialisation in similar approaches in more detail.

---

> ### Author Response · Authors · 2025-08-08
>
> - General, it would be helpful to add equation numbers to all equations.
>
> Done. All equations are now numbered.
>
> - The authors state that the paper examines the impact of hyperparameters on the _ensemble optimality gap_. However, it is never clearly stated what this optimality gap is or how it may be formalised.
>
> The *ensemble optimality gap* is now more clearly defined in both the introduction and Section 3.1
>
> - Page 5, typo: "objective: should" -> "objective: Should"
>
> Fixed. Thank you for noticing.
>
> - I think it would help to formalise the two approaches (e.g., page 7) that are evaluated against, meaning the performance of an ensemble (equation on page 2) and the average performance of individual members. The latter is clearly a less meaningful metric as it does not correspond to any sensible probabilistic model.
>
> Our intention with the "average individual performance" was to provide a clear baseline to see the raw impact of ensembling. Based on your feedback, we have revised the paper to focus the main text primarily on the final ensemble's performance, which is the key takeaway. The diagnostic metrics for individual models are now mostly relegated to the appendix to improve clarity.
>
> - I did like the results discussed in 4.1, but found that the benefits obtained through shared weight-decay to be marginal, even though the authors state _strong benefits_, which makes me think that this is overselling of the actual improvements (difference between 0.008 and 0.006 in ECE).
>
> We have toned down our claim. We now state that while the benefits can be modest given the high computational cost of joint tuning, our results do show a consistent trend: Ensembling often allows for less regularization, though the precise effect depends on the dataset, model and target metric.
>
>  - Similarly, I found the improvements in 4.2 to be modest at best, as accuracy and NLL are essentially equivalent for the ensemble, and only slight improvements in ECE can be observed.
>
> By changing the main figure, we have made it easier to compare the effects of the temperature scaling. While we don't consistently observe a meaningful improvement in calibration after temperature scaling, our results show that while individual temperature scaling is beneficial to individual models, it is not beneficial for the ensemble.
> Additionally, an important takeaway from our experiments is that the negative impacts of using a large validation set is greater than the benefits we see from temperature scaling.
>
> - Again, same story for 4.3. Here, I think the authors again oversell the improvements and even the statement that one can find "joint stopping generally outperforms disjoint holdouts" is overselling at best. At least to my eye, there is no visible improvement or difference at all.
>
> We hope that the clearer effect shown in our two new, larger-scale experiments makes this finding more convincing. We believe the trend is consistent and now better supported by the expanded empirical evidence.
>
> - I was slightly confused by the wording on page 12, saying "overall ensemble's validation performance". Is the performance showing the validation performance or the test performance in the figures of the paper? I believe it should be the latter.
>
> Model selection (e.g., determining the stopping epoch) is performed on the validation set, but all performance metrics reported in figures and tables are on the test set. We have revised the text in Section 4 to make this distinction clear.
>
> - The ablation on BatchEnsembles is somewhat ad hoc and doesn't really connect well to the rest. Unfortunately for the rest, those results seem to be the most interesting ones. I believe this section would need better integration into the rest of the paper.
> - The ablation on BatchEnsembles is somewhat too short, considering it reveals some patterns while the other ablations aren't very insightful. It would be interesting to study the effect of initialisation in similar approaches in more detail.
> - Section 4.4 needs better integration into the paper, and I would generally suggest overhauling the paper's structure to achieve better integration of the ablation.
>
> We understand this perspective. We have reframed the section to improve its connection to the rest of the paper. As parameter-efficient ensemble methods like batchensemble inherently require joint training, we wanted to investigate whether using different holdout sets was possible for such models without data leakage between ensemble members. We believe that it is an important take-away for practitioners that efficient ensembles may not always behave similarly to deep ensembles.
> We agree that it would be interesting to study the effect of initialization in related efficient ensembles, but decided against expanding the scope further to avoid detracting from the paper's main focus.

---

> ### Author Response · Authors · 2025-08-08
>
> - The results are currently rather limited, and additional ablations (using more models and datasets) would help strengthen the points.
>
> As detailed above, we have included two additional models/datasets.
>
> - As the current hyperparameter ablations show rather mild improvements at best, I believe that it would be necessary to tone down the claims in the submission and explore why those improvements are so mild.
>
> We have carefully examined our claims and revised them to reflect the modest but consistent improvements.

---

> > ### Comment · Reviewer_YiaQ · 2025-09-09
> >
> > I want to thank the authors for their efforts and responses.
> >
> > Upon reviewing the revised submission, I am pleased with the modifications and improvements made.

---

### Author Response · Authors · 2025-08-08

We thank all reviewers for their thoughtful and constructive feedback. We are pleased that the reviewers found the paper well-written, the experiments well-executed, and the topic practical and important.
Based on the feedback, we have made significant revisions to the paper:

- **Expanded Experiments:** We have doubled our experimental evaluation from two to four diverse benchmarks, adding large-scale tabular (Covertype, M=8) and text (AG News, M=8) classification tasks to our image (CIFAR-10, M=4) and graph (NCI1, M=4) experiments corresponding to thousands of additional GPU hours.
- **Clarified Methodology:** We have added a new section (3.1) to formally define the "ensemble optimality gap" and the individual vs. joint model selection strategies.
- **New Analyses:** We have added two new experiments in the appendix to address specific reviewer questions:
	- **Appendix D:** A direct comparison with the "Pool-Then-Calibrate" temperature scaling strategy from Rahaman & Thiery (2021).
	- **Appendix G:** An investigation into the interaction between joint early stopping and post-hoc temperature scaling.
- **Improved Presentation:** We have added tables with statistical comparisons to complement our figures and have revised the text to be more precise about our claims and the trade-offs involved in joint optimization.

---

### Comment · Action_Editor_c7Ek · 2025-10-27
**Please fix the bibliography**

The bibliography has many papers missing details such as venue, year, etc.  For example, "Balaji Lakshminarayanan, Alexander Pritzel, and Charles Blundell. Simple and Scalable Predictive Uncertainty Estimation using Deep Ensembles" should include NeurIPS 2017, and Deep Ensembles Work, But Are They Necessary? should include NeurIPS 2022.

Before I approve the camera ready, could you cleanup the bibliography, including adding sufficient detail.

Thank you,

Action Editor

---

> ### Author Response · Authors · 2025-10-28
>
> Dear Action Editor,
>
> Thank you for your feedback regarding the bibliography.
>
> We have now carefully reviewed and updated the entire reference list to ensure all entries include sufficient detail, such as venue and year, as requested (including Lakshminarayanan et al. and Abe et al.), along with ensuring consistency across all other references.
>
> We have uploaded the revised version.
>
> Kind regards,
>
> The Authors

---

### Decision · Action_Editor_c7Ek · 2025-09-10

**Recommendation:** Accept as is

**Additional Comments:**

The reviewers seemed quite satisfied by the changes made in the latest revision of the paper.  Therefore I'm recommending accept as-is.  Nevertheless, the reviewers pointed out a number of 'requested changes', so please do make sure to try to address these.

**Audience:**

Yes

**Audience Explanation:**

Ensembles are a very common strategy in machine learning, so any additional insights in how to make them work better are useful.

**Claims And Evidence:**

Yes

**Claims Explanation:**

The reviewers all found that the claims were substantiated by the evidence provided e.g. "In its revised form, the paper is careful, clear, and well-supported by experiments"